# MULTI-VIEW OBJECT-CENTRIC LEARNING WITH IDENTIFIABLE REPRESENTATIONS

## ABSTRACT

Modular object-centric representations are key to unlocking human-like reasoning capabilities. However, addressing challenges such as object occlusions to obtain meaningful object-level representations presents both theoretical and practical difficulties. We introduce a novel multi-view probabilistic approach that aggregates view-specific slots to capture *invariant content* information while simultaneously learning disentangled global *viewpoint-level* information. Our model resolves spatial ambiguities and provides theoretical guarantees for learning identifiable representations, setting it apart from prior work focusing on single-view settings and lacking theoretical foundations. Along with our identifiability analysis, we provide extensive empirical validation with promising results on both benchmark and proposed large-scale datasets carefully designed to evaluate multi-view methods.

## 1 INTRODUCTION

The ability to capture the notion of *objectness* in learned representations is believed to be a critical aspect for developing situation-aware AI systems with human-like *system-1* reasoning capabilities (Lake et al., 2017). Recent advances in object-centric representation learning have shown great potential in this direction (Locatello et al., 2020b; Kori et al., 2023; Löwe et al., 2024). The goal of object-centric learning (OCL) is to enable agents to learn representations of respective objects in an observed scene in the context of their environment, as opposed to learning global representations as in the case of traditional generative models such as variational auto-encoders (Kingma & Welling, 2013). Object-centric approaches enable agents to learn spatially disentangled representations, which is an important step in compositional scene generation (Bengio et al., 2013; Lake et al., 2017; Battaglia et al., 2018; Greff et al., 2020) and understanding of causal (and physical) interactions between the objects (Marcus, 2003; Gerstenberg et al., 2021; Gopnik et al., 2004).

Most of the recent progress in OCL has been limited to learning scene representations from single-viewpoints (Locatello et al., 2020b; Engelcke et al., 2021; Singh et al., 2021; Kori et al., 2023; Chang et al., 2022; Seitzer et al., 2022; Löwe et al., 2024). While these approaches may learn meaningful object-specific representations, they face insurmountable challenges due to spatial ambiguities; learning from single viewpoints cannot capture effective representations due to partially or fully occluded objects. Li et al. (2020) previously proposed an intriguing approach to address some of the spatial ambiguities. They take a view-conditional OCL perspective, which makes their approach reliant on the availability of paired viewpoint conditioning and corresponding images. Here, we take a step forward, exploring *multi-view object-centric learning* (MVOCL), allowing us to exploit objects' inherent geometry and semantics to establish correspondences across views.

Another issue with many of the earlier OCL methods (including (Li et al., 2020)) is that they lack rigorous formalisation of their underpinning explicit and implicit assumptions; Kori et al. (2024); Brady et al. (2023); Lachapelle et al. (2023) make an effort to formalise these assumptions and provide conditions under which these methods result in learning identifiable slot representations. Similarly, formalisations in MVOCL are unexplored, and the theoretical guarantees under which the partially or fully occluded slot representations are identifiable have not been studied before. In this work, we consider learning the joint distribution over all viewpoints, as opposed to view-conditional OCL (Li et al., 2020); our model provides the additional advantage of not being dependent on camera/viewpoint information. Inspired by Kori et al. (2024); Kivva et al. (2022), we take the perspective of imposing latent structure to achieve identifiable slot representations under viewpoint

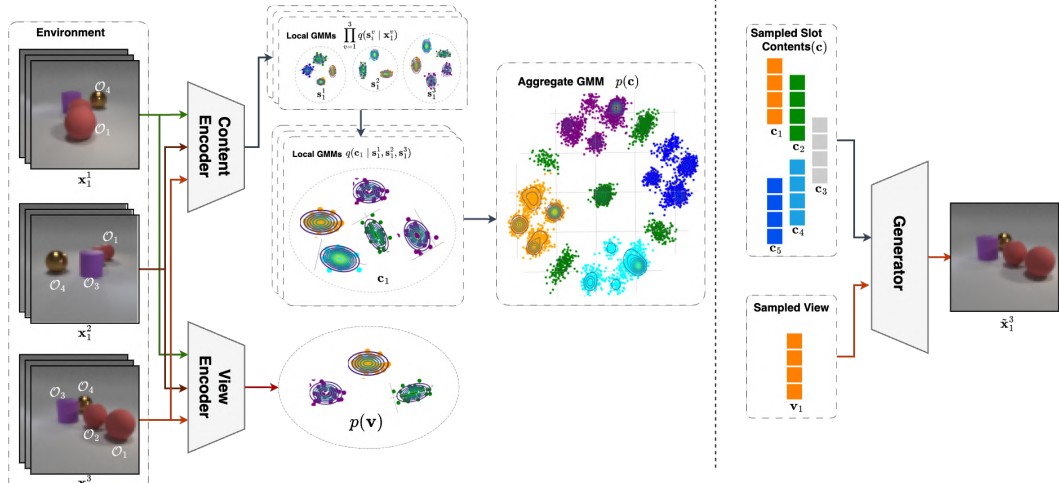

Figure 1: The figure illustrates a scene with four objects $\mathcal{O}^s = \{\mathcal{O}_1, \mathcal{O}_2, \mathcal{O}_3, \mathcal{O}_4\}$, observed from three different viewpoints, each described with a set of clearly visible objects as $\mathcal{O}^1 = \{\mathcal{O}_1, \mathcal{O}_4\}, \mathcal{O}^2 = \{\mathcal{O}_1, \mathcal{O}_3, \mathcal{O}_4\}, \mathcal{O}^3 = \{\mathcal{O}_1, \mathcal{O}_2, \mathcal{O}_3\}$. The corresponding images are passed through content and view encoders, resulting in *local* slot and global view GMMs, $q(\mathbf{s} \mid \mathbf{x})$ and $p(\mathbf{v})$, respectively. The local slot distribution is further aggregated to marginalise viewpoint information, resulting in a content GMM $q(\mathbf{c} \mid \mathbf{s})$, which is then accumulated across all samples, resulting in our optimal prior $p(\mathbf{c})$. During image generation, we sample content from $p(\mathbf{c})$ and view information from $p(\mathbf{v})$, passing them through the generator, resulting in a rendered scene from the desired viewpoint.

ambiguities. In line with Kori et al. (2024), we show that the spatial Gaussian mixture before latent distribution across viewpoints can encourage the identifiability of object-centric representations under viewpoint ambiguities without additional auxiliary data.

**Contributions:** Our main contributions in this work can be summarised as follows: (i) We propose a multi-view probabilistic slot attention MVPSA for learning identifiable object-centric representations from multiple viewpoints, resolving spatial ambiguities such as partial occlusions(§ 3); (ii) We prove that our object-centric representations are identifiable in the case of partial or full occlusions without additional view information up to an equivalence relation with a mixture model specification (§ 4); (iii) We provide conclusive empirical evidence of our identifiability results, including visual verification on synthetic 2-dimensional data; we also demonstrate the scalability of the proposed method on two new, carefully designed large-scale datasets MVMOVI-C and MVMOVI-D (§ 6). The datasets constitute a contribution on their own and are released to facilitate future work.

## 2 PRELIMINARIES

**Probabilistic Slot Attention (PSA)** as introduced by Kori et al. (2024), presents a distinct interpretation of the slot attention algorithm proposed by Locatello et al. (2020b). In PSA, a set of feature embeddings $\mathbf{z} \in \mathbb{R}^{N \times d}$ per input $\mathbf{x}$ is taken as input, and an iterative Expectation Maximization (EM) algorithm is applied over these embeddings. This process results in a Gaussian Mixture Model (GMM) characterized by mean ($\boldsymbol{\mu} \in \mathbb{R}^{K \times d}$), variance ($\boldsymbol{\sigma}^2 \in \mathbb{R}^{K \times d}$), and mixing coefficients ($\boldsymbol{\pi} \in [0, 1]^{K \times 1}$). The goal of PSA is to learn a spatial GMM for each scene, where each mean in the GMM corresponds to a specific object. In summary, PSA employs the initial mean sampled from the prior distribution and initial variance initialized with unit vector, then iteratively updates the mean based on assignment probabilities ($A_{nk}$) using Equation 2, and adjusts the variance accordingly. These updates are performed for $T$ iterations. Given that the variance is updated using closed-form updates, the objective function in the case of PSA is the negative log-likelihood of $p(\mathbf{x} \mid \boldsymbol{\mu}(T)_{1:K}, \boldsymbol{\sigma}^2_{1:K}(T), \boldsymbol{\pi}_{1:K}(T))$ for scene $\mathbf{x} \in \mathcal{X} \subseteq \mathbb{R}^{H \times W \times C}$ with $H, W, C$ corresponding to image dimensions, where the mean, variance, and mixing coefficients are updated at each iteration as described in Equation 2. Unlike slot attention (Locatello et al., 2020b), PSA learns the distribution over slots rather than just the mean where the soft assignments are determined as follows:

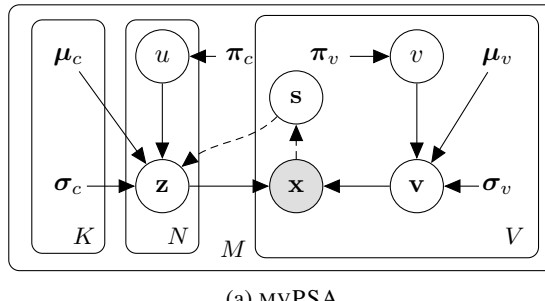 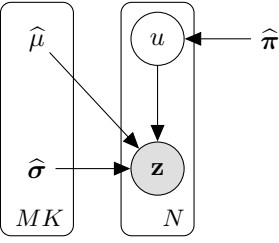

(a) MVPSA        (b) Aggregate Content Posterior

Figure 2: **Graphical model for multi-view probabilistic slot attention. (a)** MVPSA - every scene in a dataset consists of $V$ images of an environment observed from different viewpoints, with dataset $\{\{\mathbf{x}_i^v\}_{v=1}^V\}_{i=1}^M$, each image is encoded into a respective view information vector $\mathbf{v} \in \mathbb{R}^{d_v}$ resulting in a GMM distribution with $V$ components and latents $\{\mathbf{s}^v\}_{v=1}^V$, where $\mathbf{s}^v \in \mathbb{R}^{N \times d_s}$, to which a local GMM with $K$ components is fit via EM algorithm. The resulting $V$ GMM distributions are further aggregated with convex combination, marginalising the effects of view information, resulting in a view invariant content $\mathbf{c}$ GMM with $K$ components. **(b)** View invariant aggregate content distribution is obtained by marginalising out data from obtained content distribution resulting in: $q(\mathbf{c}) = \sum_{i=0}^M q(\mathbf{c} \mid \mathbf{s}, \mathbf{x})/M$. We demonstrate $q(\mathbf{c})$ and $p(\mathbf{v})$ are tractable and non-degenerate.

$$A_{nk} = \frac{\boldsymbol{\pi}(t)_k \mathcal{N}\left(\mathbf{z}_n; \boldsymbol{\mu}(t)_k, \boldsymbol{\sigma}(t)_k^2\right)}{\sum_{j=1}^K \boldsymbol{\pi}(t)_j \mathcal{N}\left(\mathbf{z}_n; \boldsymbol{\mu}(t)k, \boldsymbol{\sigma}(t)_j^2\right)}; \ \hat{A}_{nk} = A_{nk}/\sum_{l=1}^N A_{lk}; \ \boldsymbol{\pi}(t+1)_k = \sum_{n=1}^N A_{nk}/N; \quad (1)$$

$$\boldsymbol{\mu}(t+1)_k = \sum_{n=1}^N \hat{A}_{nk}\mathbf{z}_n; \ \boldsymbol{\sigma}(t+1)_k^2 = \sum_{n=1}^N \hat{A}_{nk}\left(\mathbf{z}_n - \boldsymbol{\mu}(t+1)_k\right)^2 \qquad (2)$$

**Identifiable representations.** A model is considered identifiable when different training iterations yield consistent latent distributions, thereby resulting in identical model parameters (Khemakhem et al., 2020a;c). In the context of a parameter space $\Theta$ and a family of mixing functions $\mathcal{F}$, identifiability of the model on the dataset $\mathcal{X}$ is established if, for any $\theta_1, \theta_2 \sim \Theta$ and $f_{\theta_1}, f_{\theta_2} \sim \mathcal{F}$, the condition $p(f_{\theta_1}^{-1}(\mathbf{x})) = p(f_{\theta_2}^{-1}(\mathbf{x}))$ holds for all $\mathbf{x} \in \mathcal{X}$, implying $\theta_1 = \theta_2$. However, in practical scenarios, exact equality or "strong" identifiability is often unnecessary, as establishing relationships to transformations, which can be manually recovered, proves equally effective. This concept leads to the notion of weak identifiability, where relationships are recovered up to affine transformations (Khemakhem et al., 2020c; Kivva et al., 2022). Similar identifiability relations have been elucidated for OCL in prior works (Brady et al., 2023; Lachapelle et al., 2023; Kori et al., 2024; Mansouri et al., 2023). The notion of $\sim_s$ equivalence relation is elaborated in Dfn. 1.

**Definition 1.** ($\sim_s$ equivalence (Kori et al., 2024)) Let $f_{\boldsymbol{\theta}} : \mathcal{S} \to \mathcal{X}$ denote a mapping from slot representation space $\mathcal{S}$ to image space $\mathcal{X}$ (satisfying Assumption 2), the equivalence relation $\sim_s$ w.r.t. to parameters $\boldsymbol{\theta} \in \boldsymbol{\Theta}$ is defined as: $\boldsymbol{\theta}_1 \sim_s \boldsymbol{\theta}_2 \Leftrightarrow$

$$\exists \boldsymbol{P}, \boldsymbol{H}, \mathbf{c} : f_{\boldsymbol{\theta}_1}^{-1}(\mathbf{x}; \mathbf{v}) = \boldsymbol{P}(f_{\boldsymbol{\theta}_2}^{-1}(\mathbf{x}; \mathbf{v})\boldsymbol{H} + \mathbf{a}), \forall \mathbf{x} \in \mathcal{X}, \qquad (3)$$

where $\boldsymbol{P} \in \mathcal{P} \subseteq \{0,1\}^{K \times K}$ is a permutation matrix, $\boldsymbol{H} \in \mathbb{R}^{d \times d}$ is an affine matrix, and $\mathbf{a} \in \mathbb{R}^d$.

## 3 MULTI-VIEW FORMALISM

Let $\mathbf{x}^{1:V} = \{\mathbf{x}^1, \dots \mathbf{x}^V\} \in \mathcal{X} = \mathcal{X}^1 \times \cdots \times \mathcal{X}^V$, $V$ views of the same scene observed from different viewpoints with an observational space $\mathcal{X} \subseteq \mathbb{R}^{V \times H \times W \times C}$. We consider $[V]$ as a shorthand notation for $\{1, \dots, V\}$. Let $\mathcal{O}^e = \mathcal{O}^1 \cup \cdots \cup \mathcal{O}^V$ correspond to an abstract notion of object sets of an environment, while $\mathcal{O}^v, \forall v \in [V]$ is an object set present in a considered viewpoint $v$. Importantly, we consider that the number of objects per viewpoint can vary, *i.e.*, $|\mathcal{O}^1 \cup \cdots \cup \mathcal{O}^V| \geq |\mathcal{O}^v| \ \forall \ v \in [V]$, allowing for partial or full occlusion in some viewpoints. Let $\mathbf{v}^{1:V} \in \mathcal{V} = \mathcal{V}^1 \times \cdots \times \mathcal{V}^V \subseteq \mathbb{R}^{V \times d_v}$ be inferred viewpoint-specific information[1], while $\mathbf{s}_{1:K}^{1:V} \in \mathcal{S} = \mathcal{S}^1 \times \cdots \times \mathcal{S}^V \subseteq \mathbb{R}^{V \times K \times d_s}$ correspond

---

[1]We abuse the terminology by considering viewpoint, lighting, object dimension, to be encoded in a vector $\mathbf{v}$. Note that the $\mathbf{v}$ is inferred by the model.

to a viewpoint-specific slot representation. Let $\mathbf{c}_{1:K} \in \mathcal{C} \subseteq \mathbb{R}^{K \times d_c}$ capture the notion of an *aggregate*, effectively accumulating the object knowledge across viewpoints. For any subset $A$ of $[V]$, we represent scene observations as $\mathbf{x}^A = \{\mathbf{x}^i : \forall i \in A\} \in \times_{i \in A} \mathcal{X}^i$. In this framework, the inferred viewpoints and the specific slots for each viewpoint are denoted as $\mathbf{v}^A = \{\mathbf{v}^i : \forall i \in A\} \in \times_{i \in A} \mathcal{V}^i$, and $\mathbf{s}^A_{1:K} = \{\mathbf{s}^i_{1:K} : \forall i \in A\} \in \times_{i \in A} \mathcal{S}^i$, respectively. We define $p_A(\mathbf{c})$ as the distribution of $\mathbf{c}$ over $A$. A more comprehensive summary of notations and terminologies is provided in App. A.

In modelling, without loss of generality, we consider access to a certain subset $A \subseteq [V]$, ensuring the model's applicability across different scenarios. Furthermore, to simplify notation, we sometimes do not include the superscript denoting the full set of views, thereby using $\mathbf{x} = \mathbf{x}^A$, $\mathbf{s}_{1:K} = \mathbf{s}^A_{1:K}$, and $\mathbf{v} = \mathbf{v}^A$ interchangeably. Likewise, if we do not specify the subscripts for $\mathbf{c}$ and $\mathbf{s}$, it implies they represent the entire collection of objects, specifically as $\mathbf{s} = \mathbf{s}^A_{1:K}$ and $\mathbf{c} = \mathbf{c}_{1:K}$. Lastly, given that the function $f$ operates on two distinct types of inputs, its inverse is denoted by $f^{-1}(\mathbf{x}; \mathbf{v})$, which signifies the reversal of $f$ applied to data points $\mathbf{x}$ conditioned on variable $\mathbf{v}$.

**Assumption 1.** (View-point sufficiency) For any set $A \subseteq [V]$, we consider set $A$ to be view-point sufficient iff $|\mathcal{O}^A| = |\mathcal{O}^e|$. This basically means that all the objects are visible across all the considered views $A$, even when the individual view may not contain all the object information.

---

**Example 1.** *Based on illustrated example in Figure 1, the scene is composition of four objects* $\mathcal{O}^e = \{\mathcal{O}_1, \mathcal{O}_2, \mathcal{O}_3, \mathcal{O}_4\}$, *view point subset* $A = [V] = \{1, 2, 3\}$ *is considered to be view point sufficient since* $\bigcup_{v \in A} \mathcal{O}^v = \{\mathcal{O}_1, \mathcal{O}_4\} \cup \{\mathcal{O}_1, \mathcal{O}_3, \mathcal{O}_4\} \cup \{\mathcal{O}_1, \mathcal{O}_2, \mathcal{O}_3\} = \mathcal{O}^e$.

---

**View model.** We model view as an image-level property, which we infer with the posterior $q_\theta(\mathbf{v}^v \mid \mathbf{x}^v) \forall v \in A$[2]. It is important to note that we use the same set of parameters $\theta$ across all viewpoints in $A$ for inferring view information $\mathbf{v}$. Given the access to a discrete set of viewpoints $A$, we consider prior over a view distribution to be a GMM represented by $p(\mathbf{v}) = \sum_{v=1}^{|A|} \pi_v \mathcal{N}(\mathbf{v}; \mu_v, \sigma_v^2)$.

**Viewpoint specific slots.** We extract object-level slot representations for a given image from all viewpoints; we model the slot distribution as an image conditional model described as $q(\mathbf{s}^A_{1:K} \mid \mathbf{x}^A)$, refer Figure 2a for a graphical model for the same. Similar to Probabilistic Slot Attention, we consider local GMM by fitting the individual posterior $q(\mathbf{s}^v \mid \mathbf{x}^v)$, with expectation-maximisation algorithm, resulting in the estimation of distribution parameters with closed-form updates. The resulting likelihood is described in 4, where $(\mu_i, \sigma_i^2, \pi_i)$ are mean, diagonal covariance, and mixing coefficients of an $i^{th}$ image for the considered view $v$ with $K$ components.

$$q(\mathbf{s}^A_{1:K} \mid \mathbf{x}^A_i, \mu_i, \sigma_i^2, \pi_i) = \prod_{v=1}^{|V|} \sum_{k=1}^{K} \pi_{ik} \mathcal{N}\left(\mathbf{s}^v_k; \mu_{ik}, \sigma_{ik}^2\right) \tag{4}$$

**Representation matching.** Similar to most object-centric learning approaches, the resulting view conditional slot representations are permutation invariant. To handle this invariance property, we use permutation matching function with a permutation matrix $\boldsymbol{P}$, $m_s : \mathcal{S}^A \rightarrow \mathcal{S}^A$ such that $m_s(\mathbf{s}^A_{1:K}) = \bigcup_{v=1}^{A} \boldsymbol{P} \mathbf{s}^v_{1:K}$ mapping from a given vector space to the vector space with the transformed axis. Here, we consider the content of the first Viewpoint to be the base representation and match other contents from other viewpoints to align with it. We utilise Hungarian matching, as illustrated in Locatello et al. (2020b); Emami et al. (2022); Wang et al. (2023); Kori et al. (2023), to permute object indices to align them w.r.t base representations, learning the permutation matrix $\boldsymbol{P}$.

**Content aggregator.** We consider $g : \mathcal{S} \rightarrow \mathcal{C}$ as a content aggregator function, which marginalises the effect of view conditioning. To achieve this, we align the content representations from all viewpoints $v \in A$ and perform a convex combination of these representations using the mixing coefficients of the view-specific posterior, as defined in Equation 4. Once the content representations and mixing coefficients are aligned with respect to the base representations (represented by $\tilde{\mathbf{s}}^{1:V}_{1:K}, \tilde{\boldsymbol{\pi}}^{1:V}$), the convex combination in our context accounts for potential object occlusions, which may cause objects to be absent in particular views 5. the convex combination ensures that only active representations are combined, resulting in a GMM with mixing coefficients $\boldsymbol{\pi}_k = \left(\sum_{v=1}^{|A|} \tilde{\boldsymbol{\pi}}_k)^v\right) / |A|$ and the parameters described in 6. The resulting MVPSA is illustrated in Algorithm 1.

---

[2]We consider the parametric form of $q$ to be Gaussian.

**Intuition.** Considering an example 1, given well trained model, for images $\mathbf{x}^1, \mathbf{x}^2, \mathbf{x}^2$, the resulting matched slots and mixing coefficients correspond to $\mathbf{s}^1 = \{\mathbf{s}_{\mathcal{O}_1}^1, \mathbf{s}_r^1, \mathbf{s}_r^1, \mathbf{s}_{\mathcal{O}_4}^1, \mathbf{s}_b^1\}, \mathbf{s}^2 = \{\mathbf{s}_{\mathcal{O}_3}^2, \mathbf{s}_r^2, \mathbf{s}_{\mathcal{O}_3}^2, \mathbf{s}_{\mathcal{O}_4}^2, \mathbf{s}_b^2\}, \mathbf{s}^3 = \{\mathbf{s}_{\mathcal{O}_1}^3, \mathbf{s}_{\mathcal{O}_2}^3, \mathbf{s}_{\mathcal{O}_3}^3, \mathbf{s}_r^3, \mathbf{s}_b^3\}$, where $\mathbf{s}_{\mathcal{O}_i}^v, \mathbf{s}_r^v$, and $\mathbf{s}_b^v$ correspond to slot vector for object $\mathcal{O}_i$, random slot vector and background information, respectively, with mixing coefficients $\boldsymbol{\pi}^1 = \{1, 0, 0, 1, 1\}, \boldsymbol{\pi}^2 = \{1, 0, 1, 1, 1\}$, and $\boldsymbol{\pi}^1 = \{1, 1, 1, 0, 1\}$. Proposed aggregation merges the slot information ignoring the random content vectors $\mathbf{c}_r^v$, resulting in $\mathbf{c}_{\mathcal{O}_1} = (\mathbf{s}_{\mathcal{O}_1}^1 + \mathbf{s}_{\mathcal{O}_1}^2 + \mathbf{s}_{\mathcal{O}_1}^3)/3, \mathbf{c}_{\mathcal{O}_4} = (\mathbf{s}_{\mathcal{O}_4}^1 + \mathbf{s}_{\mathcal{O}_4}^2)/2$ and so on.

$$g(\tilde{\mathbf{s}}_{1:K}^{1:V}, \tilde{\boldsymbol{\pi}}^{1:V}) = \sum_{v=1}^{|A|} \frac{\tilde{\boldsymbol{\pi}}_{1:k}^v}{\sum_{v=1}^{|A|} \tilde{\boldsymbol{\pi}}_{1:k}^v} \tilde{\mathbf{s}}_{1:K}^v; \tag{5}$$

$$\mathbb{E}(\mathbf{c}_k) = \sum_{v=1}^{|A|} \frac{\tilde{\boldsymbol{\pi}}_k^v}{\sum_{v=1}^{|A|} \tilde{\boldsymbol{\pi}}_k^v} \mathbb{E}\left(\tilde{\mathbf{s}}_k^v\right); \quad \mathbb{V}\mathrm{ar}(\mathbf{c}_k) = \sum_{v=1}^{|A|} \left(\frac{(\tilde{\boldsymbol{\pi}}_k)^v}{\sum_{v=1}^{|A|} \tilde{\boldsymbol{\pi}}_k^v}\right)^2 \mathbb{V}\mathrm{ar}\left(\tilde{\mathbf{s}}_k^v\right); \tag{6}$$

**Optimal content prior.** We rely on the fact that marginalising the effect of datapoints from posterior (*aggregate posterior*) is an optimal prior (Hoffman & Johnson, 2016; Kori et al., 2024). This results in the optimal content prior $p(\mathbf{c})$ to be an aggregate of posteriors $\iint q(\mathbf{c}|\mathbf{s}^A, \mathbf{x}^A)d\mathbf{s}^A d\mathbf{x}^A$. This imposes the structure to content distribution, rather than constraining the distribution to be close to posterior as in VAEs (Kingma & Welling, 2013), this results in the optimal prior by design, without the need for additional variational approximations. Given that GMMs are universal density approximates given enough components (even GMMs with diagonal covariances), the resulting aggregate posterior $q(\mathbf{c}) = p(\mathbf{c})$ is highly flexible and multi-modal.

**Lemma 1** (Optimal Mixture). *Given the a local content distribution $q(\mathbf{c}_{1:K} \mid \mathbf{s}_{1:K}^A, \mathbf{x}^A)$ (per-scene $\mathbf{x}^A \in \{\mathbf{x}_i^A\}_{i=1}^M$), which can be expressed as a GMM with $K|A|$ components, the aggregate posterior $q(\mathbf{c})$ is obtained by marginalizing out $\mathbf{x}, \mathbf{s}$ is a non-degenerate GMM with $MK|A|$ components:*

$$p(\mathbf{c}) = q(\mathbf{c}) = \frac{1}{M|A|} \sum_{i=1}^M \sum_{v=1}^{|A|} \sum_{k=1}^K \widehat{\boldsymbol{\pi}}_{ik} \mathcal{N}\left(\mathbf{c}; \widehat{\boldsymbol{\mu}}_{ik}, \widehat{\boldsymbol{\sigma}}_{ik}^2\right). \tag{7}$$

*Proof Sketch.* The result is obtained by integrating the product of involved latent posterior densities $q(\mathbf{c} \mid \mathbf{s}^A)q(\mathbf{s}^A \mid \mathbf{x}^A)p(\mathbf{x}^A)$. Further, we verify if the mixing coefficients sum to one in the new mixture, proving aggregated posterior to be well-defined. $\square$

**Mixing function and training objective.** In line with Kori et al. (2024), our theory does not rely on the additivity of the decoder structure; instead, we consider both additive and non-additive mixing functions denoted as $f_d : \mathcal{C} \times \mathcal{V}^v \to \mathcal{X}^v$. For additive decoders, we use a spatial-broadcasting (Greff et al., 2019) and MLP decoders, and for non-additive mixing function, we use auto-regressive transformers (Vaswani et al., 2017). We use the same network $f_d$ across all views, with trainable parameters $\theta$, which models the conditional distribution $p(\mathbf{x}^v \mid \mathbf{c}, \mathbf{v}^v)$. Probabilistically, the generative model for a view set $A$ can be described by a graphical model in figure 2a, resulting in the likelihood 8. To train our model in an end-to-end fashion, we maximise the log-likelihood of the joint distribution $p(\mathbf{x}^A)$, which results in the evidence lower bound (ELBO), Eq. 10. Here, we consider the distribution form of $p(\mathbf{x}^v \mid \mathbf{c}, \mathbf{v}^v)$ to be Gaussian with learnable mean with isotropic covariance, similarly we consider $q(\mathbf{v}^v \mid \mathbf{x})$ to be Gaussian with estimated mean and diagonal covariance.

$$p_A(\mathbf{x}^{1:V}) = \iint p_A(\mathbf{x}^{1:V} \mid \mathbf{c}_{1:K}, \mathbf{v}^{1:V}) \, p_A(\mathbf{c}_{1:K}) \, p(\mathbf{v}^{1:V}) \, d\mathbf{v}^{1:V} \, d\mathbf{c}_{1:K} \tag{8}$$

$$\log p(\mathbf{x}^A) \geq \iint q(\mathbf{v}^A \mid \mathbf{x}^A)p(\mathbf{c}_{1:K}) \log p(\mathbf{x}^A \mid \mathbf{c}_{1:K}, \mathbf{v}^A) \frac{p(\mathbf{v}_{1:K}^A)}{q(\mathbf{v}^A \mid \mathbf{x}^v)} \, d\mathbf{v}^A \, d\mathbf{c}_{1:K} \tag{9}$$

$$= \mathbb{E}_{\mathbf{c}, \mathbf{v}} \left[\log p(\mathbf{x}^A \mid \mathbf{c}, \mathbf{v})\right] - \mathrm{KL}\left(q(\mathbf{v} \mid \mathbf{x}^A) \parallel p(\mathbf{v})\right) \tag{10}$$

## 4 THEORETICAL ANALYSIS

In this section, we leverage the properties of the MVPSA method proposed in Section 3 to theoretically demonstrate the learning of identifiable representations under challenging viewpoint ambiguities. In summary, we show three main results; firstly, we show that aggregate content representations ($\mathbf{c}$) are identifiable without supervision (up to an equivalence relation). Secondly, we show that these representations are invariant to the choice of viewpoints under viewpoint sufficiency. Finally, we show that the trained model results in an approximate representational equivariance up to an affine transformation, *i.e.,* for any two viewpoints sub-sets $A, B \subseteq [V] \ni A \neq B$, the resulting content distribution $p_A(\mathbf{c})$ can be recovered by $p_B(\mathbf{c})$ up to an affine transformation.

**Assumption 2** (*Weak* Injectivity). Let $f : \mathcal{Z} \to \mathcal{X}$ be a mapping between latent space and image space, where $\dim(\mathcal{Z}) \leq \dim(\mathcal{X})$. The mapping $f_d$ is weakly injective if there exists $\mathbf{x}_0 \in \mathcal{X}$ and $\delta > 0$ such that $|f^{-1}(\{\mathbf{x}\})| = 1, \forall \mathbf{x} \in B(\mathbf{x}_0, \delta) \cap f(\mathcal{Z})$, and $\{\mathbf{x} \in \mathcal{X} : |f^{-1}(\{\mathbf{x}\})| = \infty\} \subseteq f(\mathcal{Z})$ has measure zero w.r.t. to the Lebesgue measure on $f(\mathcal{Z})$ (cf. Kivva et al. (2022)).

**Theorem 1** (Mixture of Concatenated Slots). *Let $f_s$ denote a permutation equivariant PSA function such that $f_s(\mathbf{z}^v, P\mathbf{s}^v) = P f_s(\mathbf{z}^v, \mathbf{s}^v)$, where $P \in \{0, 1\}^{K \times K}$ is an arbitrary permutation matrix. Let $\mathbf{c} = (g(\mathbf{s}_1^A, .), \ldots, g(\mathbf{s}_K^A, .)) \in \mathbb{R}^{Kd}$ be the concatenation of $K$ individual content vectors, where each vector is an aggregate of all the slots obtained from considered viewpoints in a viewpoint-set $A \subseteq [V]$ (cf. Kori et al. (2024)). Due to the nature of the aggregator function, the individual content vector is Gaussian distributed within a $K$-component mixture: $\mathbf{c}_k \sim \mathcal{N}(\boldsymbol{\mu}_k, \boldsymbol{\Sigma}_k) \in \mathbb{R}^d, \forall k \in \{1, \ldots K\}$. Then, $\mathbf{c}$ is also GMM distributed with $K!$ mixture components:*

$$p(\mathbf{c}) = \sum_{p=1}^{K!} \boldsymbol{\pi}_p \mathcal{N}(\mathbf{c}; \boldsymbol{\mu}_p, \boldsymbol{\Sigma}_p), \quad where \quad \boldsymbol{\pi} \in \Delta^{K!-1}, \quad \boldsymbol{\mu}_p \in \mathbb{R}^{Kd}, \quad \boldsymbol{\Sigma}_p \in \mathbb{R}^{Kd \times Kd}. \quad (11)$$

**Theorem 2.** *(Affine Equivalence of aggregate content) For any subset $A \subseteq [V]$, such that $|A| > 0$, given a set of images $\mathbf{x}^A \in \mathcal{X}^A$ and a corresponding aggregate content $\mathbf{c} \in \mathcal{C}$ and a non-degenerate content posterior $q(\mathbf{c} \mid \mathbf{s}^A)$, considering two mixing function $f_d, \tilde{f}_d$ satisfying assumption 2, with a shared image, then $\mathbf{c}$ are identifiable up to $\sim_s$ equivalence.*

> **Intuition.** Considering an example 1, with two perfectly trained models $f_d$ and $\tilde{f}_d$. Resulting aggregate contents are described as $\mathbf{c} = f_d^{-1}(\mathbf{x}^A; \mathbf{v}^A) = \{\mathbf{c}_{\mathcal{O}_1}, \mathbf{c}_{\mathcal{O}_2}, \mathbf{c}_{\mathcal{O}_3}, \mathbf{c}_{\mathcal{O}_4}, \mathbf{c}_{\mathcal{O}_b}\}$ and $\tilde{\mathbf{c}} = \tilde{f}_d^{-1}(\mathbf{x}^A; \mathbf{v}^A) = \{\tilde{\mathbf{c}}_{\mathcal{O}_2}, \tilde{\mathbf{c}}_{\mathcal{O}_4}, \tilde{\mathbf{c}}_{\mathcal{O}_3}, \tilde{\mathbf{c}}_{\mathcal{O}_1}, \tilde{\mathbf{c}}_{\mathcal{O}_b}\}$ for $A = [V] = \{1, 2, 3\}$. $\sim_s$ equivalence states that there exists a permutation matrix $P$ which aligns the object order in $\tilde{\mathbf{c}}$ to match with $\mathbf{c}$ and there exists and invertible affine mapping $A$ such that $\tilde{\mathbf{c}}_{\mathcal{O}_k} = A\mathbf{c}_{\mathcal{O}_k} \forall k \in \{1, 2, 3, 4\}$.

*Proof Sketch.* To prove the following result, we follow multiple steps as described below: (i). We demonstrate the distribution $p(\mathbf{c})$ obtained as a result of lemma 1 is non-degenerate and a valid distribution, (ii). With the above results, we demonstrate invertibility restrictions on mixing functions, (iii). Finally, we constrain the subspace to affine, demonstrating $\sim_s$ of aggregate content $\mathbf{c}$. ☐

**Theorem 3.** *(Invariance of aggregate content) For any subset $A, B \subseteq [V]$, such that $|A| > 0, |B| > 0$ and both $A, B$ satisfy an assumption 1, we consider aggregate content to be invariant to viewpoints if $f_A \sim_s f_B$ for data $\mathcal{X}^A \times \mathcal{X}^B$.*

> **Intuition.** Considering an example 1, with $A = \{1, 3\}, B = \{2, 3\}$, such that both sets $A, B$ are viewpoint sufficient. Let $f_A$ and $f_B$, correspond to perfectly trained models on $\mathcal{X}^A$ and $\mathcal{X}^B$ respectively. Resulting aggregate slots are described as $\mathbf{c} = f_A^{-1}(\mathbf{x}^A; \mathbf{v}^A) = \{\mathbf{c}_{\mathcal{O}_1}, \mathbf{c}_{\mathcal{O}_2}, \mathbf{c}_{\mathcal{O}_3}, \mathbf{c}_{\mathcal{O}_4}, \mathbf{c}_{\mathcal{O}_b}\}$ and $\tilde{\mathbf{c}} = \tilde{f}_B^{-1}(\mathbf{x}^B; \mathbf{v}^B) = \{\tilde{\mathbf{c}}_{\mathcal{O}_2}, \tilde{\mathbf{c}}_{\mathcal{O}_4}, \tilde{\mathbf{c}}_{\mathcal{O}_3}, \tilde{\mathbf{c}}_{\mathcal{O}_1}, \tilde{\mathbf{c}}_{\mathcal{O}_b}\}$. Content invariance states that there exists a permutation matrix $P$ which aligns the object order in $\tilde{\mathbf{c}}$ to match with $\mathbf{c}$, and there exists an invertible affine mapping $A$ such that $\tilde{\mathbf{c}}_{\mathcal{O}_k} = A\mathbf{c}_{\mathcal{O}_k}$, even when the model is trained on completely different scenes with same objects.

*Proof Sketch.* To prove the following result, we extend the proof of Theorem 2, and first establish that there exist two inevitable affine functions $h_A, h_B$ for mixing functions $f_A, f_B : \mathcal{C} \times \mathcal{V} \to \mathcal{X}$ to

map representations $\mathbf{c}$ with a given view set $\mathbf{v}^A$ to observations $\mathbf{x}^A$. Later, we show that, in the case of invariance, an affine mapping exists from $h_A$ to $h_B$. □

**Theorem 4.** *(Approximate representational equivariance) For a given aggregate content $\mathbf{c}$, for any two views $\mathbf{v}, \tilde{\mathbf{v}} \sim p_A(\mathbf{v})$, resulting in respective scenes $\mathbf{x} \sim p_A(\mathbf{x} \mid \mathbf{v}, \mathbf{c})$ and $\tilde{\mathbf{x}} \sim p_A(\mathbf{x} \mid \tilde{\mathbf{v}}, \mathbf{c})$, for any homeomorphic transformation $h_x \in \mathcal{H}_x$ such that $h_x(\mathbf{x}) = \tilde{\mathbf{x}}$, their exists another homeomorphic transformation $h_v \in \mathcal{H}_v$ such that $\mathcal{H}_v \subseteq \mathcal{H}_x \subseteq \mathbb{R}^{\dim(\mathbf{x})}$ and $\mathbf{v} = h_v^{-1}\left(f_d^{-1}(h_x(\mathbf{x}); \mathbf{c})\right)$.*

*Remark* 1. Note that we do not claim viewpoint equivariance here. Instead, we say that the transformation function transforming the view representations $\mathbf{v}$ as an effect of the homeomorphic transformation of $\mathbf{x}$ lies in the same subspace of input transformations.

*Remark* 2. Implications of this result: the homography matrix $\boldsymbol{H}$ between two cameras with non-degenerate relative pose matrix, with fixed intrinsic camera matrices and non-zero translation and rotation matrix is a homeomorphic transformation (Hartley & Zisserman, 2003).

> **Intuition.** In the scenario when the cameras are positioned such that they have overlapping fields of view, and their relative pose (rotation and translation) must avoid degeneracies like aligning on the same plane or mapping points to infinity. This results in the transformation between views being smooth, invertible, and consistent. If the scene is planar or depth variations are minimal, the homography can capture the transformation accurately without the need for inverse rendering. Notably, the cameras should have non-zero rotation and translation to avoid collapsing the scene, and their intrinsic parameters must be known or identical to prevent distortions. When the scenario satisfies all the above properties, the 2D homography transformation $\boldsymbol{H}$ between two camera views can be learned as a homeomorphic transformation.

*Proof Sketch.* We prove the following result by following the steps in theorem 3, over a view distribution $p(\mathbf{v})$ but for a fixed content vector $\mathbf{c}$. □

## 5  RELATED WORKS

**Identifiable representation learning.** Learning meaningful representations from unlabeled data has long been a primary objective of deep learning (Bengio et al., 2013). Several approaches, such as those proposed by (Higgins et al., 2017; Kim & Mnih, 2018; Eastwood & Williams, 2018; Mathieu et al., 2019), relied on independence assumptions between latent variables to learn disentangled representations. However, Hyvärinen & Pajunen (1999); Locatello et al. (2019) demonstrated the provable impossibility of unsupervised methods for learning independent latent representations from i.i.d. data. Which is tackled by restricting mixing functions to conformal maps (Buchholz et al., 2022) or volume-preserving transformations (Yang et al., 2022), or with additional data assumptions (Zimmermann et al., 2021; Locatello et al., 2020a; Brehmer et al., 2022; Ahuja et al., 2022; Von Kügelgen et al., 2021), or by imposing structure in the latent space as in nonlinear Independent Component Analysis (ICA) (Hyvarinen et al., 2019; Khemakhem et al., 2020a;b), resulting in identifiable models. In the context of nonlinear ICA, Dilokthanakul et al. (2016) introduced a VAE model with a GMM prior, and Willetts & Paige (2021) empirically demonstrated the effectiveness of the GMM prior, which was later rigorously proven by Kivva et al. (2022). Kori et al. (2024) use this notion of latent GMM in the context of OCL, achieving identifiability guarantees for object-centric representations. Here, we use this notion in the context of multiview object-centric representations, tackling the issues with spatial ambiguities and uncertainties in bindings.

**Multiview nonlinear ICA.** It has been noted that addressing the challenge of nonlinear Independent Component Analysis (ICA) can involve incorporating a learnable clustering task within the latent representations, thereby imposing asymmetry in the latent distribution (Willetts & Paige, 2021; Kivva et al., 2022). Moreover, the study by Gresele et al. (2020) delves into multiview nonlinear ICA, particularly in scenarios involving corrupted observations, where they aim to recover invariant representations while accounting for certain ambiguities. Along similar lines, Daunhawer et al. (2023); Von Kügelgen et al. (2021) explore the concept of style-content identification using contrastive learning, focusing on addressing the multiview nonlinear ICA problem. Here, we work along similar lines by emphasising the learning of invariant content and identifiable object-centric representations. We achieve this by formulating a reconstruction objective where the enforced invariance and equivariance stem from the underlying probabilistic graphical model rather than relying on a contrastive learning

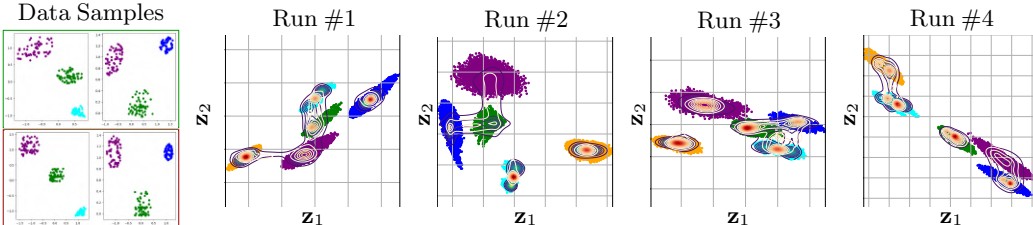

Figure 3: **Identifiability of** $q(\mathbf{c})$ **and** $q(\mathbf{s})$. First image illustrates 2 datapoints observed from 2 different viewpoints enclosed in green and red boxes, respectively). Recovered marginalised slot distribution ($q(\mathbf{s})$–blue contours) and marginalised content distribution ($q(\mathbf{c})$–orange contours, across 4 runs of MVPSA. As detailed in CASE STUDY 1, we used a 2D synthetic dataset with 5 total 'objects', with each observation containing at most 3. This provides strong evidence of recovery of the latent space up to affine transformations, empirically verifying our claims in Theorem 2.

Table 1: Comparing identifiability of $q(\mathbf{s})$, $q(\mathbf{c})$, and $p(\mathbf{v})$ scores wrt existing OCL methods.

| METHOD | CLEVR-AUG | | | CLEVR-MV | | | GQN | | |
|---|---|---|---|---|---|---|---|---|---|
| | SMCC ↑ | INV-SMCC ↑ | MCC ↑ | SMCC ↑ | INV-SMCC ↑ | MCC ↑ | SMCC ↑ | INV-SMCC ↑ | MCC ↑ |
| AE | 0.26 ± .01 | - | 0.26 ± .02 | 0.32 ± .02 | - | 0.29 ± .02 | 0.29 ± .02 | - | 0.22 ± .02 |
| SA | 0.45 ± .05 | - | 0.28 ± .02 | 0.47 ± .03 | - | 0.29 ± .01 | 0.38 ± .02 | - | 0.29 ± .01 |
| PSA | 0.48 ± .03 | - | 0.28 ± .01 | 0.49 ± .02 | - | 0.32 ± .02 | 0.38 ± .02 | - | 0.29 ± .01 |
| MulMON | 0.56 ± .04 | 0.57 ± .01 | - | 0.61 ± .03 | 0.62 ± .02 | - | 0.61 ± .03 | 0.62 ± .02 | - |
| **MVPSA** | 0.64 ± .01 | 0.66 ± .01 | 0.63 ± .04 | 0.67 ± .01 | 0.66 ± .01 | 0.69 ± .04 | 0.59 ± .01 | 0.63 ± .01 | 0.52 ± .03 |

objective. Similar to the noiseless setting in Gresele et al. (2020), we demonstrate the recovery of invariant content representations using different subsets of viewpoints.

**Object-centric learning.** Extending nonlinear ICA from representation learning to object-specific representational learning has been heavily explored before (Burgess et al., 2019; Engelcke et al., 2019; Greff et al., 2019) by employing an iterative variational inference approach (Marino et al., 2018), whereas Van Steenkiste et al. (2020); Lin et al. (2020) adopt more of a generative perspective, studied the effect of object binding and scene composition empirically. Recently, the use of iterative attention mechanisms has gained a significant interest (Locatello et al., 2020b; Engelcke et al., 2021; Singh et al., 2021; Wang et al., 2023; Singh et al., 2022; Emami et al., 2022). Most of these works operate in a single-view setting, which causes fundamental issues of viewpoint ambiguities in terms of occlusions and uncertainties in binding. Recent methods including Eslami et al. (2018); Arsalan Soltani et al. (2017); Tobin et al. (2019); Wu et al. (2016) consider single object from multiple views to tackle this particular problem, additionally Kosiorek et al. (2018); Hsieh et al. (2018); Li et al. (2020) explore multi-object binding in videos and multiple views, tackling object binding issues across frames. Despite their empirical effectiveness, most of these works lack formal identifiability guarantees. In line with recent efforts analysing theoretical guarantees in object-centric representations (Lachapelle et al., 2023; Brady et al., 2023; Kori et al., 2024), we formally investigate the modelling assumptions and their implications for achieving identifiability guarantees in the context of multi-object, multiview object-centric representation learning settings.

## 6 EMPIRICAL EVALUATION

Given the work's theoretical focus, experimentally, we aim to provide strong empirical evidence of our identifiability, invariance, and equivariance claims in a multiview setting. We also extend our experiments to standard imaging benchmarks along with large-scale images with high variability, demonstrating the framework's scalability and applicability in high-dimensional settings.

**Experimental setup.** We consider standard benchmark datasets from OCL literature, including CLEVR-MV, CLEVR-AUG, GQN (Li et al., 2020), and proposed datasets MV-MOVIC, MV-MOVID which are multiview versions of MoViC dataset with fixed and scene-specific cameras (Greff et al., 2022). To verify our claims on (i) identifiability claim, we train our model on a given view subset $A \subseteq [V]$ and compare view averaged SMCC measures as described in Kori et al. (2024), (ii) invariance claim, we train multiple models on different subsets of viewpoints $A, B \subseteq [V]$ and compare the aggregate content representations across models, quantifying the similarities with SMCC, we consider

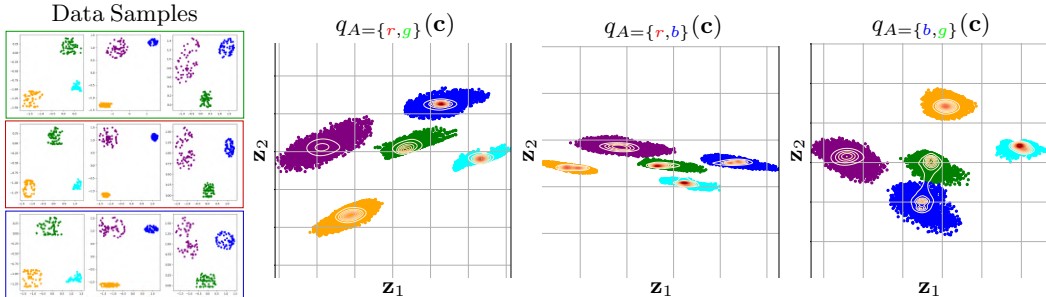

Figure 4: **Viewpoint invariance for** $q(\mathbf{c})$. First image illustrates 3 datapoints observed from 3 different viewpoints enclosed in green, red, blue boxes, respectively). Recovered marginalised aggregate content distribution $q(\mathbf{c})$ when trained with different view pairs {(green, red), (red, blue), (green, blue)} are illustrated in later figures. As the resulting distributions with different datasets only vary by an affine transformation, providing strong evidence for Theorem 3.

this measure to be invariant SMCC (INV-SMCC), and finally, (iii) for subspace equivariance, we consider a trained model with a view subset $A \subseteq [V]$ and compute MCC of view information $\mathbf{v}$ by applying random homeomorphic transformations on samples $\mathbf{x}^A \sim \mathcal{X}^A$ (which can also be done by considering samples $\mathbf{x}^B \sim \mathcal{X}^B$, where cameras relative position satisfy the required constraints 2).

**Models & baselines.** We consider two ablations with two types of decoders: (i) additive with MLPs and spatial broadcasting CNNs and (ii) non-additive decoders, which include transformer models. In all cases, we use LeakyReLU activations to satisfy the weak injectivity conditions (Assumption 2). In terms of object-centric learning baselines, we compare with standard additive autoencoder setups following (Brady et al., 2023), slot-attention (SA) (Locatello et al., 2020b), probabilistic slot-attention (PSA) (Kori et al., 2024), and MulMON (Li et al., 2020).

CASE STUDY 1: ILLUSTRATION OF IDENTIFIABILITY RESULTS. To definitively show the validity of our claims about identifiability (Theorem 2, Theorem 3, and Theorem 4), we created a synthetic scenario for modeling. This setup enables us to visually examine both the aggregate posterior distributions and the prior distributions in detail. The process used for generating data is thoroughly explained in App. C.1. In Figure 3, we display the distributions of marginalized slots and the aggregate content distribution $q(\mathbf{s})$ and $q(\mathbf{c})$, comparing different runs that are either rotated, skewed, or mirrored with respect to each other. To quantitatively measure the same, we computed SMCC and observed it to be $\mathbf{0.95} \pm 0.01$, empirically verifying our Theorem 2. Furthermore, to illustrate the invariance of distribution $q(\mathbf{c})$ across viewpoints (Theorem 3), we consider three different viewpoints. We use all possible pairs to learn $q(\mathbf{c})$ distributions as illustrated in Figure 4, where the distributions from second to last sub-figures are learned wrt viewpoints described by {g, r}, {r, b}, and {g, b}, respectively. Similar to our previous findings, these distributions were also found to be rotated, skewed, or mirrored relative to each other, with an observed SMCC of $\mathbf{0.87} \pm 0.11$, further confirming the claims in Theorem 3.

CASE STUDY 2: IMAGING APPLICATIONS. In this section, we demonstrate the generalizability and scalability of our method to higher-dimensional image settings. We first evaluate the framework on synthetic benchmarks, specifically focusing on CLEVR-MV, CLEVR-AUG, and GQN with simple objects. Given the *true generative factors* are unobserved, we derive our quantitative assessments from multiple runs. The results are shown in Table 1, confirming the validity of our theory on imaging datasets. Regarding the baseline comparisons that utilize a single viewpoint, the INV-SMCC mirrors the SMCC due to its inherent design (*i.e.,* aggregation of a set with a single element is the same element). Moreover, in the case of MULMON, the model does not estimate view information, but use the observed view conditioning, rendering the MCC metric inapplicable. Figure 5 showcases how the number of viewpoints impacts the identifiability of the $\mathbf{s}$, $\mathbf{v}$, and $\mathbf{c}$ variables; the involved experiments reflect the increase in performance with an increase in the number of views to a certain extent, across all three benchmark datasets.

Additionally, we applied our methodology to our proposed MV-MOVIC and MV-MOVID datasets. The latter enables us to examine how the model performs when the assumption detailed in 1 is not satisfied. To evaluate model behaviour in an environment with consistant objects but with different

Table 2: Identifiability and generalisability analysis on MV-MOVIC dataset.

| METHOD | IN-DOMAIN RESULTS | | | | OUT-OF-DOMAIN RESULTS | | | |
|---|---|---|---|---|---|---|---|---|
| | mBO ↑ | SMCC ↑ | INV-SMCC ↑ | MCC ↑ | mBO ↑ | SMCC ↑ | INV-SMCC ↑ | MCC ↑ |
| SA-MLP | 0.28 ± 0.091 | 0.36 ± 0.004 | - | 0.38 ± 0.004 | 0.26 ± 0.08 | 0.38 ± 0.006 | - | 0.43 ± 0.016 |
| PSA-MLP | 0.30 ± 0.022 | 0.38 ± 0.002 | - | 0.43 ± 0.012 | 0.30 ± 0.03 | 0.40 ± 0.005 | - | 0.43 ± 0.019 |
| MVPSA-MLP | 0.28 ± 0.021 | 0.52 ± 0.021 | 0.61 ± 0.023 | 0.54 ± 0.026 | 0.27 ± 0.02 | 0.51 ± 0.029 | 0.58 ± 0.031 | 0.52 ± 0.021 |
| SA-TRANSFORMER | 0.34 ± 0.014 | 0.36 ± 0.016 | - | 0.46 ± 0.009 | 0.33 ± 0.041 | 0.36 ± 0.043 | - | 0.45 ± 0.008 |
| PSA-TRANSFORMER | 0.37 ± 0.021 | 0.38 ± 0.007 | - | 0.47 ± 0.007 | 0.37 ± 0.033 | 0.39 ± 0.016 | - | 0.45 ± 0.008 |
| MVPSA-TRANSFORMER | 0.38 ± 0.008 | 0.44 ± 0.003 | 0.46 ± 0.001 | 0.53 ± 0.011 | 0.36 ± 0.017 | 0.46 ± 0.033 | 0.46 ± 0.018 | 0.55 ± 0.082 |

scenarios, we conducted in-domain and out-of-domain (OOD) evaluations. For in-domain analysis, the model is trained and assessed on the same viewpoint group $A = [1, 2, 3]$. Conversely, for OOD evaluation, we consider the previously trained model, but test it against a new set of viewpoints $B = [3, 4, 5]$. The findings presented in Table 2 regarding the MV-MOVIC dataset reveal that the SMCC, INV-SMCC, and MCC metrics show similar performance across both domains. This indicates that the distributional characteristics remain unchanged when both the training and testing environments contain the same objects. The MV-MOVID dataset analysis can be found in App. F.

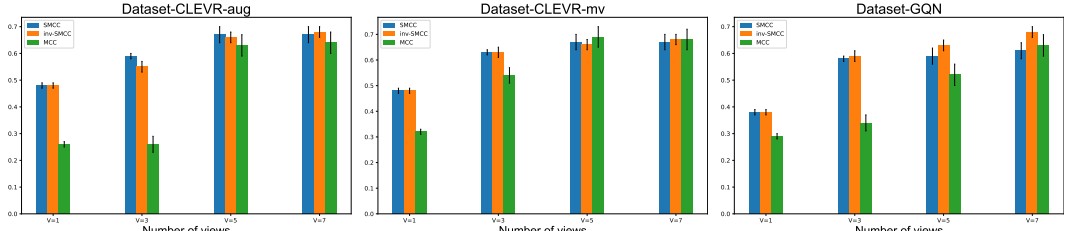

Figure 5: Influence of Number of viewpoints on identifiability for synthetic datasets.

# 7 CONCLUSION & DISCUSSION

Understanding when object-centric representations are both unambiguous and identifiable is essential for developing large-scale models with provable correctness guarantees. Unlike most existing work on identifiability, which largely focuses on single-view setups, we offer identifiability guarantees in multi-view scenarios. Building upon the approach by Kori et al. (2024), we use distributional assumptions for latent slot and view representations, drawing inspiration from mixture model-based structures. To achieve this, we propose a model that is viewpoint-agnostic and does not require additional view-conditioning information.

Our model specifically guarantees the identifiability of view-specific slot representations, viewpoint-invariant content representations, and view representations, all without the need for additional supervision (up to an equivalence relation). We visually validate our theoretical claims using illustrative 2D data points. We then empirically demonstrate the model's identifiability properties on multiple object-centric benchmarks, highlighting its ability to resolve view ambiguities in imaging applications. Furthermore, we showcase the scalability of our approach on large-scale datasets and more complex decoders using realistic datasets and transformer decoders, respectively, demonstrating its capacity to scale effectively with both data volume and decoder complexity.

**Limitations & future work.** We recognize that our assumptions, particularly regarding the *viewpoint sufficiency*, are strong and may not always hold in practice. However, we did not observe limiting effects of this assumption on the proposed MV-MOVID dataset. A more extensive analysis of this assumption and its implications in real-world applications is left for future work. We would also highlight that the *weak injectivity* of the mixing function may not always hold for different types of architectures. While generally applicable, the piecewise-affine functions we use may not always capture valid assumptions for real-world problems, *e.g.*, when the model is misspecified. Nevertheless, to the best of our knowledge, our theoretical results on multi-object, multi-view identifiability are unique and capture key concepts in multi-view object-centric representation learning, opening various new avenues for future research.

## REPRODUCIBILITY STATEMENT

To ensure the reproducibility of our research, we will be making all relevant code, data, and documentation available. The benchmark datasets used are publicly available, and for the additionally proposed datasets, the data-generating scripts and the datasets themselves are provided with instructions for further research. We detail all the involved hyper-parameters later in the appendix, along with hardware requirements to reproduce our results.

## BROADER STATEMENT

This paper proposes a multi-view probabilistic slot attention algorithm, addressing spatial ambiguities to achieve identifiable object-centric representations. The work extends theoretical advancements in the field of OCL, and as such it has little immediate societal or ethical consequences. Our method might be a step towards interpretable, equivariant, and aligned models, which are desired properties of trustworthy AI.

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

# A    NOTATIONS

| | |
|---|---|
| $\mathcal{O}^v$ | : Abstract object set as observed from viewpoint $v$. |
| $[V] = \{1, \dots, V\}$ | : Exhaustive set of viewpoints, representing all possible views. |
| $A, B \subset [V]$ | : Subsets of viewpoints, selecting specific views from the complete set. |
| $\mathcal{X} = \times_{v \in A} \mathcal{X}^v$ | : Data space, formed by the Cartesian product of data spaces for each view in subset $A$. |
| $\mathbf{x}^A = \{\mathbf{x}^v : \forall v \in A\} \sim \mathcal{X}$ | : Data sample, where $\mathbf{x}^v$ is the data from view $v$, and $\mathbf{x}^A$ represents the set of data across all views in $A$. |
| $f_e$ | : Encoder model, which maps input data to a latent space or feature representation. |
| $\mathbf{z}$ | : Spatial latent features, representing inferred spatial properties from the data across views. |
| $\mathcal{S}$ | : View-specific slot space, a space for features that are tied to particular viewpoints. |
| $\mathcal{C}$ | : View-invariant content space, representing features that are constant across different viewpoints. |
| $\mathbf{s} \sim \mathcal{S}$ | : Samples from the view-specific slot space, representing view-dependent latent features. |
| $\mathbf{c} \sim \mathcal{C}$ | : Samples from the view-invariant content space, representing features that remain consistent across views. |
| $f_s, \tilde{f}_s$ | : Probabilistic slot attention module, responsible for attending to and disentangling different parts of the input related to different views. |
| $f_d, \tilde{f}_d$ | : Mixing function, which combines view-specific and view-invariant features into a unified representation. |
| $\mathcal{V}$ | : View information space, a space that encodes information specific to each viewpoint (e.g., angle, position). |
| $\mathbf{v} \sim \mathcal{V}$ | : A sample from the view information space representing a specific view or camera configuration. |
| $f_v, \tilde{f}_v$ | : View extractor function, which extracts viewpoint-related information from the data. |
| $\boldsymbol{\mu}_c, \boldsymbol{\mu}_s, \boldsymbol{\mu}_v$ | : Mean of invariant content, view-specific slots, and view distributions. |
| $\boldsymbol{\sigma}_c, \boldsymbol{\sigma}_s, \boldsymbol{\sigma}_v$ | : Standard deviation of invariant content, view-specific slots, and view distributions. |
| $\boldsymbol{\pi}_c, \boldsymbol{\pi}_s, \boldsymbol{\pi}_v$ | : Mixing coefficients of invariant content, view-specific slots, and view distributions. |
| $A_{nk}$ | : Assignment confidence of a slot $k$ getting mapped to token $n$. |
| $\boldsymbol{P} \in \mathcal{P} \subseteq \{0,1\}^{K \times K}$ | : Permutation matrix. |
| $m_s$ | : Matching function used to align object representations across views. |
| $\Delta^K$ | : Simplex in the space of dimension K. |
| $\mathcal{H}_x, \mathcal{H}_v$ | : Space of homeomorphic transformation. |

## B  ALGORITHM

Here we illustrate all the steps involved in the of proposed method MVPSA, refer 1.

---

**Algorithm 1** Multi-view Probabilistic Slot Attention MVPSA

---

1: **Input:** $A \in [V]$, $\mathbf{z}^A = \{f_e(\mathbf{x}^v) \, \forall v \in A\} \in \mathbb{R}^{|A| \times N \times d}$        ▷ input representations
2: $\mathrm{key}^A \leftarrow \boldsymbol{W}_k \mathbf{z}^A \in \mathbb{R}^{|A| \times N \times d}$, $\mathrm{value}^A \leftarrow \boldsymbol{W}_v \mathbf{z}^A \in \mathbb{R}^{|A| \times N \times d}$    ▷ optional value := key
3: $\mathbf{s} \leftarrow \emptyset$; $\hat{\boldsymbol{\pi}} \leftarrow \emptyset$
4: **for** $v \in A$ **do**
5:     $\forall k, \boldsymbol{\pi}(0)_k \leftarrow 1/K$, $\boldsymbol{\mu}(0)_k \sim \mathcal{N}(0, \mathbf{I}_d)$, $\boldsymbol{\sigma}(0)_k^2 \leftarrow \mathbf{1}_d$
6:     **for** $t = 0 \rightarrow T-1$ **do**
7:        $A_{nk} \leftarrow \frac{\boldsymbol{\pi}(t)_k \mathcal{N}\left(\mathrm{key}_n; \boldsymbol{W}_q \boldsymbol{\mu}(t)_k, \boldsymbol{\sigma}(t)_k^2\right)}{\sum_{j=1}^{K} \boldsymbol{\pi}(t)_j \mathcal{N}\left(\mathrm{key}_n; \boldsymbol{W}_q \boldsymbol{\mu}(t)_j, \boldsymbol{\sigma}(t)_j^2\right)}$        ▷ compute attention
8:        $\hat{A}_{nk} \leftarrow \frac{A_{nk}}{\sum_{l=1}^{N} A_{lk}}$        ▷ normalize attention
9:        $\boldsymbol{\mu}(t+1)_k \leftarrow \sum_{n=1}^{N} \hat{A}_{nk} \cdot \mathrm{value}_n$        ▷ update slot mean
10:       $\boldsymbol{\sigma}(t+1)_k^2 \leftarrow \sum_{n=1}^{N} \hat{A}_{nk} \cdot (\mathrm{value}_n - \boldsymbol{\mu}(t+1)_k)^2$       ▷ update slot variance
11:       $\boldsymbol{\pi}(t+1)_k \leftarrow \frac{1}{N} \sum_{n=1}^{N} A_{nk}$       ▷ update mixing coefficient
12:     **end for**
13:     $\mathbf{s} \leftarrow \mathbf{s} \cup \{(\boldsymbol{\mu}_{1:K}(T), \boldsymbol{\sigma}_{1:K}^2(T))\}$; $\hat{\boldsymbol{\pi}} \leftarrow \hat{\boldsymbol{\pi}} \cup \{\boldsymbol{\pi}_{1:K}(T)\}$        ▷ slot collection
14: **end for**
15: **return** ConvexCombination$(\mathbf{s}, \hat{\boldsymbol{\pi}})$        ▷ $K$ view invariant content

---

## C  DATASETS

### C.1  ILLUSTRATIVE DATASET

To visually illustrate the effectiveness of our theory we experiment with 2 dimensional illustrative dataset. For this, similar to Kori et al. (2024), we defined a $K = 5$ component GMM, with differing mean parameters $\boldsymbol{\mu} = \{\boldsymbol{\mu}_1, \ldots, \boldsymbol{\mu}_5\}$, and shared isotropic covariances, which we use to sample locations for object. For a given location we randomly select one object from 'cube', 'cylinder', 'torus', 'pyramid', and 'sphere' and generate 64 random points on the surface of the selected shape uniformly covering it. To create a single data point, we randomly select three of the five locations and place a randomly selected object at the location. To include multiple viewpoints, we consider $V = 2$ camera location and project the objects creating two different scenes. We use different colors representing different objects in Figure 3, 4 and used 1000 data points in total to train our toy MVPSA models.

### C.2  PROPOSED DATASET

In this work, we introduce the MV-MOVI datasets, created using Kubric Greff et al. (2022), which feature multi-view scenes with segmentation annotations. We propose two variants of the dataset: MV-MOVIC, where the camera locations for every viewpoint remain fixed across all scenes, and MV-MOVID, where the camera locations dynamically change for each scene.

Both MV-MOVIC and MV-MOVID primarily consist of scenes generated by randomly selecting a background from a set of **458** available options and choosing $K$ objects, where $3 \leq K \leq 6$, from a pool of **930** objects. In total, the datasets contain **3,000** scenes, each captured from 5 different viewpoints. Additionally, each scene has 24 frames of data and object segmentation masks for every frames are provided for all 5 views to facilitate the evaluation of model performance.

## D  MASK GENERATION

In the case of additive decoders, the decoder outputs $K$ three channelled tensors along with $K$ single channelled mask. We consider normalise these masks with softmax transformation along

slot dimension, ensuring the each pixel only contribute to a single slot. The resulting softmaxed masks are used in composing (image $= \sum_k \text{mask}_k \cdot \text{image}_k$) the slots to reconstruct an image for training. During inference we normalise masks with `sigmoid` transformation, allowing us to estimate occluded objects visually resolving the spatial ambiguities, with occluded objects. In later section, we illustrate the results with both `softmax` and `sigmoid` transformations.

## D.1 ADDITIVITY IMPLICATIONS

As pointed out in Lachapelle et al. (2023), `softmax`-based masks do not truly fall under the category of additive decoders due to the competition between masks for groups of pixels. This implies that the additive decoders studied in Lachapelle et al. (2023) are not expressive enough to represent the "masked decoders" typically employed in object-centric representation learning. The issue arises from the normalization of alpha masks, and care must be taken when extrapolating the findings from Lachapelle et al. (2023) to the models used in practice.

Although `sigmoid`-based masks satisfy the condition of additivity during inference, it is important to note that in our setting the model is still trained using `softmax` normalization. The effect of using `sigmoid` masks during inference can be visually observed in Appendix F.

## E IDENTIFIABILITY PROOFS

**Lemma 1.** (Optimal Content Mixture) For $A \in [V]$, given the a local content distribution $q(\mathbf{c}_{1:K} \mid \mathbf{s}_{1:K}^A, \mathbf{x}^A)$ (per-scene $\mathbf{x}^A \in \{\mathbf{x}_i^A\}_{i=1}^M$), which can be expressed as a GMM with $K$ components, the aggregate posterior $q(\mathbf{c})$ is obtained by marginalizing out $\mathbf{x}, \mathbf{s}$ is a non-degenerate global Gaussian mixture with $MK|A|$ components:

$$p(\mathbf{c}) = q(\mathbf{c}) = \frac{1}{M|A|} \sum_{i=1}^M \sum_{v=1}^{|A|} \sum_{k=1}^K \widehat{\boldsymbol{\pi}}_{ik} \mathcal{N}\left(\mathbf{c}; \widehat{\boldsymbol{\mu}}_{ik}, \widehat{\boldsymbol{\sigma}}_{ik}^2\right). \tag{12}$$

*Proof.* We begin by noting that the aggregate posterior $q(\mathbf{c})$ is the optimal prior $p(\mathbf{c})$ so long as our posterior approximation $q(\mathbf{c} \mid \mathbf{s}^A, \mathbf{x}^A)$ is close enough to the true posterior $p(\mathbf{c} \mid \mathbf{s}^A, \mathbf{x}^A)$, since for a dataset $\mathbf{x}^A \in \{\mathbf{x}_i^A\}_{i=1}^M$, for which we start with $q(\mathbf{s}^A \mid \mathbf{x}^A)$, we have that:

$$p(\mathbf{s}^A) = \int p(\mathbf{s}^A \mid \mathbf{x}^A) p(\mathbf{x}^A) d\mathbf{x}^A \tag{13}$$

$$= \mathbb{E}_{\mathbf{x}^A \sim p(\mathbf{x}^A)} \left[ p(\mathbf{s}^A \mid \mathbf{x}^A) \right] \tag{14}$$

$$\approx \frac{1}{M} \sum_{i=1}^M p(\mathbf{s}^A \mid \mathbf{x}_i^A) \tag{15}$$

$$\approx \frac{1}{M} \sum_{i=1}^M q(\mathbf{s}^A \mid \mathbf{x}_i^A) \tag{16}$$

$$=: q(\mathbf{s}^A), \tag{17}$$

We further extend this to $q(\mathbf{c})$ as follows

$$p(\mathbf{c}) = \int p(\mathbf{c} \mid \mathbf{s}^A) p(\mathbf{s}^A) d\mathbf{s}^A \tag{18}$$

$$= \mathbb{E}_{\mathbf{s}^A \sim p(\mathbf{s}^A)} \left[ p(\mathbf{c} \mid \mathbf{s}^A) \right] \tag{19}$$

$$\approx \frac{1}{M} \sum_{i=1}^M p(\mathbf{c} \mid \mathbf{s}_i^A) \tag{20}$$

$$\approx \frac{1}{M} \sum_{i=1}^M q(\mathbf{c} \mid \mathbf{s}_i^A) \tag{21}$$

$$=: q(\mathbf{c}), \tag{22}$$

where we approximated $p(\mathbf{x})$ using the empirical distribution, then substituted in the approximate posterior and marginalized out $\mathbf{x}$ to get $p(\mathbf{s})$, we later consider the distributional form of $p(\mathbf{s})$ and marginalise $\mathbf{s}^A$ to get $p(\mathbf{c})$. This observation was first made by Hoffman & Johnson (2016) and was used in Kori et al. (2024) we use it to motivate our setup. Given PSA fits a local GMM to each latent variable sampled from the approximate posterior: $\mathbf{z}^A \sim q(\mathbf{z}^A \mid \mathbf{x}_i^A)$, for $i = 1, \ldots, M$. Let $f_s(\mathbf{z}^A)$ denote the (local) the product of GMM density, its expectation is given by:

$$\mathbb{E}_{p(\mathbf{x}^A), q(\mathbf{z}^A \mid \mathbf{x}^A)} \left[ f_s(\mathbf{z}^A) \right] = \iint p(\mathbf{x}^A) q(\mathbf{z}^A \mid \mathbf{x}^A) f_s(\mathbf{z}^A) d\mathbf{x}^A d\mathbf{z}^A \tag{23}$$

$$\approx \iint \frac{1}{M} \sum_{i=1}^{M} \delta(\mathbf{x}^A - \mathbf{x}_i^A) q(\mathbf{z}^A \mid \mathbf{x}^A) f(\mathbf{z}) d\mathbf{x}^A d\mathbf{z}^A \tag{24}$$

$$= \int \frac{1}{M} \sum_{i=1}^{M} q(\mathbf{z}^A \mid \mathbf{x}_i^A) f(\mathbf{z}^A) d\mathbf{z}^A \tag{25}$$

$$= \int \frac{1}{M} \sum_{i=1}^{M} \mathcal{N}\left(\mathbf{z}^A; \boldsymbol{\mu}(\mathbf{x}_i^A), \boldsymbol{\sigma}^2(\mathbf{x}_i^A)\right) \cdot$$

$$\sum_{k=1}^{K} \boldsymbol{\pi}_k(\mathbf{x}_i^A) \mathcal{N}\left(\mathbf{z}^A; \boldsymbol{\mu}_k(\mathbf{x}_i^A), \boldsymbol{\sigma}_k^2(\mathbf{x}_i^A)\right) d\mathbf{z}^A \tag{26}$$

$$\approx \int \frac{1}{M} \sum_{i=1}^{M} \delta(\mathbf{z}^A - \boldsymbol{\mu}(\mathbf{x}_i^A)) \cdot \sum_{k=1}^{K} \boldsymbol{\pi}_k(\mathbf{x}_i^A) \mathcal{N}\left(\mathbf{z}^A; \boldsymbol{\mu}_k(\mathbf{x}_i^A), \boldsymbol{\sigma}_k^2(\mathbf{x}_i^A)\right) d\mathbf{z}^A \tag{27}$$

$$= \frac{1}{M} \sum_{i=1}^{M} \sum_{k=1}^{K} \boldsymbol{\pi}_k(\mathbf{x}_i^A) \mathcal{N}\left(\mathbf{z}^A; \boldsymbol{\mu}_k(\mathbf{x}_i^A), \boldsymbol{\sigma}_k^2(\mathbf{x}_i^A)\right) \tag{28}$$

$$=: q(\mathbf{z}^A), \tag{29}$$

where we again used the empirical distribution approximation of $p(\mathbf{x})$, and the following basic identity of the Dirac delta to simplify: $\int \delta(\mathbf{x} - \mathbf{x}') f_e(\mathbf{x}) d\mathbf{x} = f_e(\mathbf{x}')$.

For the general case, however, we must instead compute the product of $q(\mathbf{z}^A \mid \mathbf{x}^A)$ and $f_s(\mathbf{z}^A)$ rather than use a Dirac delta approximation as in Equation 27. To that end we may proceed as follows w.r.t. to each datapoint $\mathbf{x}_i^A$:

$$q(\mathbf{z}^A \mid \mathbf{x}_i^A) \cdot f_s(\mathbf{z}^A) = \mathcal{N}\left(\mathbf{z}^A; \boldsymbol{\mu}(\mathbf{x}_i^A), \boldsymbol{\sigma}^2(\mathbf{x}_i)\right) \cdot \sum_{k=1}^{K} \boldsymbol{\pi}_k(\mathbf{x}_i^A) \mathcal{N}\left(\mathbf{z}^A; \boldsymbol{\mu}_k(\mathbf{x}_i^A), \boldsymbol{\sigma}_k^2(\mathbf{x}_i^A)\right) \tag{30}$$

$$= \sum_{k=1}^{K} \boldsymbol{\pi}_k(\mathbf{x}_i^A) \left[ \mathcal{N}\left(\mathbf{z}^A; \boldsymbol{\mu}(\mathbf{x}_i^A), \boldsymbol{\sigma}^2(\mathbf{x}_i)\right) \cdot \mathcal{N}\left(\mathbf{z}^A; \boldsymbol{\mu}_k(\mathbf{x}_i^A), \boldsymbol{\sigma}_k^2(\mathbf{x}_i^A)\right) \right] \tag{31}$$

$$= \sum_{k=1}^{K} \sum_{v=1}^{|A|} \widehat{\boldsymbol{\pi}}_{ivk} \mathcal{N}\left(\mathbf{z}; \widehat{\boldsymbol{\mu}}_{ivk}, \widehat{\boldsymbol{\sigma}}_{ivk}^2\right), \tag{32}$$

Given the product of GMM is a GMM with the number of components equal to product of number of components in individual GMM, however in our setting we consider all the components in individual GMM across viewpoints are aligned resulting in GMM with number of compoenents equal to sum of individual components which in our case correspond to $|A|K$. The posterior parameters of the resulting mixture are given in closed-form by:

$$\widehat{\boldsymbol{\sigma}}_{ivk}^2 = \left( \frac{1}{\boldsymbol{\sigma}_k^2(\mathbf{x}_i^v)} + \frac{1}{\boldsymbol{\sigma}^2(\mathbf{x}_i^v)} \right)^{-1}, \qquad \widehat{\boldsymbol{\mu}}_{ivk} = \widehat{\boldsymbol{\sigma}}_{ivk}^2 \left( \frac{\boldsymbol{\mu}(\mathbf{x}_i^v)}{\boldsymbol{\sigma}^2(\mathbf{x}_i^v)} + \frac{\boldsymbol{\mu}_k(\mathbf{x}_i^v)}{\boldsymbol{\sigma}_k^2(\mathbf{x}_i^v)} \right), \tag{33}$$

which are the standard distributional parameters obtained from a product of two Gaussians.

Now to show that the resulting GMM is non-degenerate we need to show $\sum_{k=1}^{K} \widehat{\pi}_{ivk} = 1$, for $i = 1, 2, \ldots, M, v \in A$. Based on equation 28:

$$\implies \frac{1}{M|A|} \sum_{i=1}^{M} \sum_{k=1}^{K} \widehat{\pi}_{ivk} = \frac{1}{M|A|} \sum_{i=1}^{M} 1 = \frac{1}{M|A|} \cdot M|A| = 1, \tag{34}$$

$$\implies \frac{1}{M|A|} \sum_{i=1}^{M} \sum_{k=1}^{K} \widehat{\pi}_{ivk} = 1. \tag{35}$$

based on the above equation we can say that the scaled sum of the mixing proportions of all $K$ components in all $M$ GMMs along all $|A|$ views when the components are aligned must equal 1, show that the resulting aggregate posterior is non-degenerate and a valid probability distribution. $\square$

We additionally borrow some theorems and definitions from Kivva et al. (2022) which are essential for our proofs. First, we restate the definition of a *generic point* as outlined by Kivva et al. (2022) below.

**Definition 2.** A point $\mathbf{x} \in f_d(\mathbb{R}^m) \subseteq \mathbb{R}^n$ is generic if there exists $\delta > 0$, such that $f_d : B(\mathbf{s}, \delta) \to \mathbb{R}^n$ is affine for every $\mathbf{s} \in f_d^{-1}(\{\mathbf{x}\})$

**Theorem 5** (Kivva et al. Kivva et al. (2022)). *Given $f_d : \mathbb{R}^m \to \mathbb{R}^n$ is a piecewise affine function such that $\{\mathbf{x} \in \mathbb{R}^n : |f_d^{-1}(\{\mathbf{x}\})| = \infty\} \subseteq f_d(\mathbb{R}^m)$ has measure zero with respect to the Lebesgue measure on $f_d(\mathbb{R}^m)$, this implies $\dim(f_d(\mathbb{R}^m)) = m$ and almost every point in $f_d(\mathbb{R}^m)$ (with respect to the Lebesgue measure on $f_d(\mathbb{R}^m)$) is generic with respect to $f_d$.*

**Theorem 6** (Kivva et al. Kivva et al. (2022)). *Consider a pair of finite GMMs in $\mathbb{R}^m$:*

$$\mathbf{y} = \sum_{j=1}^{J} \boldsymbol{\pi}_j \mathcal{N}(\mathbf{y}; \boldsymbol{\mu}_j, \boldsymbol{\Sigma}_j), \qquad and \qquad \mathbf{y}' = \sum_{j=1}^{J'} \boldsymbol{\pi}'_j \mathcal{N}(\mathbf{y}'; \boldsymbol{\mu}'_j, \boldsymbol{\Sigma}'_j). \tag{36}$$

*Assume that there exists a ball $B(\mathbf{x}, \delta)$ such that $\mathbf{y}$ and $\mathbf{y}'$ induce the same measure on $B(\mathbf{x}, \delta)$. Then $\mathbf{y} \equiv \mathbf{y}'$, and for some permutation $\tau$ we have that $\boldsymbol{\pi}_i = \boldsymbol{\pi}'_{\tau(i)}$ and $(\boldsymbol{\mu}_i, \boldsymbol{\Sigma}_i) = (\boldsymbol{\mu}'_{\tau(i)}, \boldsymbol{\Sigma}'_{\tau(i)})$.*

**Theorem 7** (Kivva et al. Kivva et al. (2022)). *Given $\mathbf{z} \sim \sum_{i=1}^{J} \boldsymbol{\pi}_i \mathcal{N}(\mathbf{z}; \boldsymbol{\mu}_i, \boldsymbol{\Sigma}_i)$ and $\mathbf{z}' \sim \sum_{j=1}^{J'} \boldsymbol{\pi}'_j \mathcal{N}(\mathbf{z}'; \boldsymbol{\mu}'_j, \boldsymbol{\Sigma}'_j)$ and $f_d(\mathbf{z})$ and $\tilde{f}_d(\mathbf{z}')$ are equally distributed. We can assume for $\mathbf{x} \in \mathbb{R}^n$ and $\delta > 0$, $f_d$ is invertible on $B(\mathbf{x}, 2\delta) \cap f_d(\mathbb{R}^m)$. This implies that there exists $\mathbf{x}_1 \in B(\mathbf{x}, \delta)$ and $\delta_1 > 0$ such that both $f_d$ and $\tilde{f}_d$ are invertible on $B(\mathbf{x}_1, \delta_1) \cap f_d(\mathbb{R}^m)$.*

**Theorem 2** (Affine Equivalence of aggregate content) For any subset $A \subseteq [V]$, such that $|A| > 0$, given a set of images $\mathbf{x}^A \in \mathcal{X}^A$ and a corresponding aggregate content $\mathbf{c} \in \mathcal{C}$ and a non-degenerate content posterior $q(\mathbf{c} \mid \mathbf{s}^A)$, considering two mixing function $f_d, \tilde{f}_d$ satisfying assumption 2, with a shared image, then $\mathbf{c}$ are identifiable up to $\sim_s$ equivalence.

*Proof.* Based on the results of Kori et al. (2024) we know that when $p(\mathbf{s})$ is aggregate posterior of $q(\mathbf{s} \mid \mathbf{x})$, $p(\mathbf{s}$ is identifiable upto $\sim_s$ equivalence. Additionally, based on lemma 1 we know that both $q(\mathbf{s} \mid \mathbf{x})$ and $q(\mathbf{c} \mid \mathbf{s})$ are a non-degenerate GMM with valid probability distribution. Using similar arguments in Kori et al. (2024); Kivva et al. (2022) we show that $p(\mathbf{c})$ and $p(\mathbf{s})$ are identifiable up to $\sim_s$ equivalence.

We know that

$$p(\mathbf{s}^A) = \int q(\mathbf{s}_{1:K}^A \mid \mathbf{x}^A) p(\mathbf{x}^A) d\mathbf{x}^A \tag{37}$$

$$= \int \prod_{v \in A} q(\mathbf{s}^v \mid \mathbf{x}^v) p(\mathbf{x}^v) d\mathbf{x}^A \tag{38}$$

$$= \int \prod_{v \in A} \left( \sum_{k=1}^{K} \boldsymbol{\pi}_k \mathcal{N}\left(\mathbf{s}^v; \boldsymbol{\mu}_k(\mathbf{x}^v), \boldsymbol{\sigma}_k^2(\mathbf{x}^v)\right) \right) p(\mathbf{x}^v) d\mathbf{x}^A \tag{39}$$

$$= \prod_{v \in A} \frac{1}{|\mathcal{X}|} \int \left( \sum_{k=1}^{K} \boldsymbol{\pi}_k \mathcal{N}\left(\mathbf{c}^v; \boldsymbol{\mu}_k(\mathbf{x}^v), \boldsymbol{\sigma}_k^2(\mathbf{x}^v)\right) \right) \delta(\mathbf{x}^v - \mathbf{x}_i^v) d\mathbf{x}^A \tag{40}$$

$$= \prod_{v \in A} \left( \sum_{k=1}^{|\mathcal{X}|K} \frac{1}{|\mathcal{X}|} \tilde{\boldsymbol{\pi}}_{vk} \mathcal{N}\left(\mathbf{s}^v; \tilde{\boldsymbol{\mu}}_{vk}, \tilde{\boldsymbol{\sigma}}_{vk}^2\right) \right) \tag{41}$$

Change of variables from $\mathbf{s}$ to $\mathbf{c}$ to get prior over random variable $\mathbf{c}$, with matching function $g$, results in:

$$p(\mathbf{c}_{1:K}) = \int p(\mathbf{s}_{1:K}^A) \delta\left(\mathbf{s}_{1:K}^A - g(\mathbf{s}_{1:K}^A, \boldsymbol{\pi}_{A,1:K})\right) d\mathbf{c}_{1:K}^A \tag{42}$$

Given the transformation $g$ is linear, resulting us with the distribution with mean given by:

$$\mathbb{E}_{\mathbf{c}}\left(\mathbf{c}_{1:K}\right) = \mathbb{E}_{\mathbf{s}}\left(g(\mathbf{s}_{1:K}^A, \boldsymbol{\pi}_{A,1:K})\right) \tag{43}$$

$$= g\left(\mathbb{E}_{\mathbf{s}}(\mathbf{s}_{1:K}^A), \boldsymbol{\pi}_{A,1:K}\right) \tag{44}$$

$$= \sum_{u \in A} \frac{\boldsymbol{\pi}_{u,1:K}}{\sum_{u \in A} \boldsymbol{\pi}_{u,1:K}} \mathbb{E}_{\mathbf{s}}(\mathbf{s}_{1:K}^A) \tag{45}$$

and the covariance follows the diagonal structure as in $p(\mathbf{c})$, which can be described as follows:

$$\mathbb{V}\mathrm{ar}(\mathbf{c}_{1:K}) = \sum_{v \in A} \left( \frac{\boldsymbol{\pi}_{v,1:K}}{\sum_{v \in A} \boldsymbol{\pi}_{v,1:K}} \right)^2 \mathbb{V}\mathrm{ar}_{\mathbf{c}}(\mathbf{c}_{1:K}^A) \tag{46}$$

Finally, the mixture components can be expressed as:

$$\tilde{\boldsymbol{\pi}}_{1:K} = \frac{\sum_{v \in A} \tilde{\boldsymbol{\pi}}_{v,1:K}}{|A|} \tag{47}$$

With distribution parameters described in equations 45, 46, and 47, we define the aggregate content distribution as GMM expressed as follows:

$$p(\mathbf{c}) = \sum_{k=1}^{|\mathcal{X}|K} \frac{1}{|\mathcal{X}|} \frac{\sum_{v \in A} \tilde{\boldsymbol{\pi}}_{vk}}{|A|} \mathcal{N}\left( \mathbf{v}; \sum_{v \in A} \frac{\tilde{\boldsymbol{\pi}}_{vk}}{\sum_{v \in A} \tilde{\boldsymbol{\pi}}_{vk}} \tilde{\boldsymbol{\mu}}_{vk}, \sum_{v \in A} \left( \frac{\tilde{\boldsymbol{\pi}}_{vk}}{\sum_{v \in A} \tilde{\boldsymbol{\pi}}_{vk}} \right)^2 \tilde{\boldsymbol{\sigma}}_{vk}^2 \right) \tag{48}$$

**Validity of $p(\mathbf{c})$:** The outer summation in equation 48 can be split into two one for image samples and other for original mixing coefficients, which results in the equation:

$$p(\mathbf{c}) = \sum_{i=1}^{|\mathcal{X}|} \sum_{k=1}^{K} \frac{1}{|\mathcal{X}|} \frac{\sum_{v \in A} \tilde{\boldsymbol{\pi}}_{vik}}{|A|} \mathcal{N}\left( \mathbf{c}; \sum_{v \in A} \frac{\tilde{\boldsymbol{\pi}}_{vik}}{\sum_{v \in A} \tilde{\boldsymbol{\pi}}_{vik}} \tilde{\boldsymbol{\mu}}_{vik}, \sum_{v \in A} \left( \frac{\tilde{\boldsymbol{\pi}}_{vik}}{\sum_{v \in A} \tilde{\boldsymbol{\pi}}_{vik}} \right)^2 \tilde{\boldsymbol{\sigma}}_{vik}^2 \right) \tag{49}$$

Based on this we can observe the each component in our GMM corresponds to particular slots for a given image in a given viewpoint, triple describing each component is:

$$\left\{ \tilde{\boldsymbol{\pi}}_{vik}, \tilde{\boldsymbol{\mu}}_{vik}, \tilde{\boldsymbol{\sigma}}_{vik}^2 \right\}, \quad \text{for} \quad v = 1, \dots, |A| \quad i = 1, 2, \dots, |\mathcal{X}|, \quad \text{and} \quad k = 1, 2, \dots, K. \tag{50}$$

To verify that $p(\mathbf{c})$ is a non-degenerate mixture, we observe the following implication:

$$\sum_{i=1}^{|\mathcal{X}|}\sum_{k=1}^{K}\frac{1}{|\mathcal{X}|}\frac{\sum_{v\in A}\tilde{\boldsymbol{\pi}}_{vik}}{|A|}=1, \tag{51}$$

$$\implies \frac{1}{|\mathcal{X}|}\frac{1}{|A|}\sum_{i=1}^{|\mathcal{X}|}\sum_{v\in A}\sum_{k=1}^{K}\tilde{\boldsymbol{\pi}}_{vik}=\frac{1}{|\mathcal{X}|}\frac{1}{|A|}|\mathcal{X}|\cdot|A|\cdot1=1 \tag{52}$$

similar to lemma 1, this says that the scaled sum of the mixing proportions of all $K$ components in all $|\mathcal{X}|$ GMMs must equal 1, proving that the associated aggregate posterior mixture $p(\mathbf{c})$ is a well-defined and non degenerate probability distribution.

**Invertibility restrictions:**  Given two piece-wise affine compositional functions $f_d, \tilde{f}_d : \mathcal{C}\times\mathcal{V} \to \mathcal{X}$, for a given set of views $\mathbf{v}^A$, let $\mathbf{c}=(\mathbf{c}_1,\ldots,\mathbf{c}_K), \ni \mathbf{c}_k \sim \mathcal{N}(\mathbf{c}_k; \boldsymbol{\mu}_k, \boldsymbol{\Sigma}_k)$ and $\mathbf{c}'=(\mathbf{c}'_1,\ldots,\mathbf{c}'_K), \ni \mathbf{c}'_k \sim \mathcal{N}(\mathbf{c}'_k; \boldsymbol{\mu}'_k, \boldsymbol{\Sigma}'_k)$ be a pair of aggregate content representations, result of sampling a concatenated higher dimensional GMM distribution in $\mathbb{R}^{Kd}$, as shown in Theorem 1, Kori et al. (2024). In the case when, $f_{d\sharp}(\mathcal{C}, \{\mathbf{v}^A\})$ and $\tilde{f}_{d\sharp}(\mathcal{C}', \{\mathbf{v}^A\})$[3] are equally distributed. Now assume that there exists $\mathbf{x}^A \in \mathcal{X}$ and $\delta > 0$ such that $f_d$ and $\tilde{f}_d$ are invertible and piecewise affine on $B(\mathbf{x}^A, \delta) \cap f_d(\mathcal{S})$, for a given set of views $\mathbf{v}^A$, which implies $\dim f_d(\mathcal{C}, \{\mathbf{v}^A\}) = |\mathcal{C}|$.

**Affine subspace:**  We now restrict the space $B(\mathbf{x}^A, \delta)$ to a subspace $B(\mathbf{x}'^A, \delta')$ where $\mathbf{x}^A \in B(\mathbf{x}'^A, \delta')$ such that $f_d$ and $\tilde{f}_d$ are now invertible and affine on $B(\mathbf{x}'^A, \delta') \cap f_d(\mathcal{C} \times \{\mathbf{v}^A\})$. With $L \subseteq \mathcal{X}^A$ be an $|\mathcal{C}|$-dimensional affine subspace (assuming $|\mathcal{X}^A| \geq |\mathcal{C}|$), such that $B(\mathbf{x}'^A, \delta') \cap f_{d\sharp}(\mathcal{C}, \{\mathbf{v}^A\}) = B(\mathbf{x}'^A, \delta') \cap L$. We also define $h_f, h_{\tilde{f}} : \mathcal{C} \to L$ to be a pair of invertible affine functions where $h_{f\sharp}^{-1}(B(\mathbf{x}'^A, \delta') \cap L) = f_{d\sharp}^{-1}(B(\mathbf{x}'^A, \delta') \cap L; \mathbf{v}^A)$ and $h_{\tilde{f}\sharp}^{-1}(B(\mathbf{x}'^A, \delta') \cap L) = \tilde{f}_{d\sharp}^{-1}(B(\mathbf{x}'^A, \delta') \cap L; \mathbf{v}^A)$. Implying $h_f(\mathbf{c})$ and $h_{\tilde{f}}(\mathbf{c}')$ are finite GMMs that coincide with $B(\mathbf{x}'^A, \delta') \cap L$ and $h_f(\mathbf{c}) \equiv h_{\tilde{f}}(\mathbf{c}')$, theorem 6, Kivva et al. (2022). Given, $h = h_{\tilde{f}}^{-1} \circ h_f$ and $h_f(\mathbf{c})$ and $h_{\tilde{f}}(\mathbf{c}')$ then $h$ is an affine transformation such that $h(\mathbf{c}) = \mathbf{c}'$.

$\sim_s$ **equivalence:**  Given Theorems 5 and 7, there exists a point $\mathbf{x} \in f_{d\sharp}(\mathcal{C}, \{\mathbf{v}^A\})$ that is generic with respect $f_d$ and $\tilde{f}_d$ and invertible on $B(\mathbf{x}, \delta) \cap f_{d\sharp}(\mathcal{C}, \{\mathbf{v}^A\})$. Having established that there is an affine transformation $h(\mathbf{c}) = \mathbf{c}'$ and invertiblility of piece-wise affine functions $f_d$ and $\tilde{f}_d$ on $B(\mathbf{x}^A, \delta) \cap f_{d\sharp}(\mathcal{C}, \{\mathbf{v}^A\})$, this implies that $\mathbf{c}$ is identifiable up to an affine transformation and permutation of $\mathbf{c}_k \in \mathbf{c}$, concluding our proof.

**Remark:**  Given Theorem 6, we know that for each higher dimensional mixture component in $p(\mathbf{c})$ induces the same measure on $B(\mathbf{x}^A, \delta)$ and hence for some permutation $\tau$ we have that $(\boldsymbol{\mu}_{\pi(i)}, \boldsymbol{\Sigma}_{\pi(i)}) = (\boldsymbol{\mu}'_{\tau(\pi(i))}, \boldsymbol{\Sigma}'_{\tau(\pi(i))})$. Therefore, each mixture component $\mathbf{c}_{\pi(i)}$ is identifiable up to affine transformation, and permutation of aggregate content representations in $\mathbf{c}$. Now, given sampling $\mathbf{c}_k$ is equivalent to obtaining $K$ samples from the GMM, $q(\mathbf{z})$ and concatenating, this makes $q(\mathbf{z})$ identifiable up to affine transformation, $h$ and permutation of slot representations in $\mathbf{c}$. It now trivially follows that each of the aggregate content representation $\mathbf{c}_k \sim \mathcal{N}(\mathbf{c}_k; \boldsymbol{\mu}_k, \boldsymbol{\Sigma}_k) \in \mathbb{R}^d, \forall k \in \{1, \ldots, K\}$ is identifiable up to affine transformation, $h$ based on the following observed property of GMMs:

$$\sum_{k=1}^{K}\boldsymbol{\pi}_k h_\sharp\left(\mathcal{N}(\mathbf{s}_k; \boldsymbol{\mu}_k, \boldsymbol{\Sigma}_k)\right) \sim h_\sharp\left(\sum_{k=1}^{K}\boldsymbol{\pi}_k\mathcal{N}(\mathbf{s}'_k; \boldsymbol{\mu}'_k, \boldsymbol{\Sigma}'_k)\right), \tag{53}$$

$$\square$$

**Theorem 3**  (Invariance of aggregate content) For any subset $A, B \subseteq [V]$, such that $|A| > 0, |B| > 0$ and both $A, B$ satisfy an assumption 1, we consider aggregate content to be invariant to viewpoints if $f_A \sim_s f_B$ for data $\mathcal{X}^A \times \mathcal{X}^B$.

---

[3]$f_{d\sharp}$ correspond to push forward operation, applying function $f_d$ on all the elements of the given set.

*Proof.* Based on equation 48, $p_A(\mathbf{s})$ and $p_B(\mathbf{s})$ can be expressed as follows:

$$p_A(\mathbf{c}) = \sum_{k=1}^{|\mathcal{X}|K} \frac{1}{|\mathcal{X}|} \frac{\sum_{v \in A} \tilde{\boldsymbol{\pi}}_{vk}}{|A|} \mathcal{N}\left(\mathbf{c}; \sum_{v \in A} \frac{\tilde{\boldsymbol{\pi}}_{vk}}{\sum_{v \in A} \tilde{\boldsymbol{\pi}}_{vk}} \tilde{\boldsymbol{\mu}}_{vk}, \sum_{v \in A} \left(\frac{\tilde{\boldsymbol{\pi}}_{vk}}{\sum_{v \in A} \tilde{\boldsymbol{\pi}}_{vk}}\right)^2 \tilde{\boldsymbol{\sigma}}_{vk}^2\right) \quad (54)$$

$$p_B(\mathbf{c}) = \sum_{k=1}^{|\mathcal{X}|K} \frac{1}{|\mathcal{X}|} \frac{\sum_{u \in B} \tilde{\boldsymbol{\pi}}_{uk}}{|B|} \mathcal{N}\left(\mathbf{c}; \sum_{u \in B} \frac{\tilde{\boldsymbol{\pi}}_{uk}}{\sum_{u \in B} \tilde{\boldsymbol{\pi}}_{uk}} \tilde{\boldsymbol{\mu}}_{uk}, \sum_{u \in B} \left(\frac{\tilde{\boldsymbol{\pi}}_{uk}}{\sum_{u \in B} \tilde{\boldsymbol{\pi}}_{uk}}\right)^2 \tilde{\boldsymbol{\sigma}}_{uk}^2\right) \quad (55)$$

Given the assumption of viewpoint sufficiency 1 we know the objects observed in viewpoint set $A$ are same as the object observed in set $B$. Following the results of Theorem 2, we know that both $p_A(\mathbf{s})$ and $p_B(\mathbf{s})$ are independently identifiable up to $\sim_s$ equivalence, which means $f_A$ and $f_B$ are invertible for a given views $\mathbf{v}^A$ and $\mathbf{v}^B$ respectively.

**Affine mapping.** Without loss of generality, for a given set of views $\mathbf{v}^A$, there exists some $L \subseteq \mathcal{X}^A$ be an $|\mathcal{S}|$-dimensional affine subspace, such that $B(\mathbf{x}'^A, \delta) \cap f_{A\sharp}(\mathcal{C}, \{\mathbf{v}^A\}) \cap f_{B\sharp}(\mathcal{C}, \{\mathbf{v}^A\}) = B(\mathbf{x}'^A, \delta) \cap L$. This implies their exists an affine map between $\mathbf{c} = f_A^{-1}(\mathbf{x}^A; \mathbf{v}^A)$ and $\tilde{\mathbf{c}} = f_B^{-1}(\mathbf{x}^B; \mathbf{v}^A)$. Let $h_A : \mathcal{C} \to L$ to be an invertible affine functions where $h_{A\sharp}^{-1}(B(\mathbf{x}'^A, \delta') \cap L) = f_{A\sharp}^{-1}(B(\mathbf{x}'^A, \delta') \cap L; \mathbf{v}^A) = f_{B\sharp}^{-1}(B(\mathbf{x}'^B, \delta') \cap L; \mathbf{v}^A)$ resulting in $h_A(\mathbf{c}) = \mathbf{c}'$. Similarly, we can show their exists an affine map between $\tilde{\mathbf{c}} = f_A^{-1}(\mathbf{x}^A; \mathbf{v}^B)$ and $\tilde{\mathbf{c}}' = f_B^{-1}(\mathbf{x}^B; \mathbf{v}^B)$, such that $h_B(\tilde{\mathbf{c}}) = \tilde{\mathbf{c}}'$.

**Invariance setup.** In the case when representations are invariant, $p_A(\mathbf{c})$ and $p_B(\mathbf{c})$ are equally distributed, which means aggregate content domain in both cases are same or similar $\mathcal{C}_A = \mathcal{C}_B$.

$$\mathbf{c}' = h(\tilde{\mathbf{c}}') \quad (56)$$

$$\implies h_A(\mathbf{c}) = (h \circ h_B)(\tilde{\mathbf{c}}) \quad (57)$$

$$\implies \mathbf{c} = (h_A^{-1} \circ h \circ h_B)(\tilde{\mathbf{c}}) \quad (58)$$

Given composition of affine maps is affine, we can consider the mapping $(h_A^{-1} \circ h \circ h_B)$ to be an affine, resulting in an $\sim_s$ equivalence between $f_A$ and $f_B$.

$\square$

**Theorem 4** (Approximate representational equivariance) For a given aggregate content $\mathbf{c}$, for any two views $\mathbf{v}, \tilde{\mathbf{v}} \sim p_A(\mathbf{v})$, resulting in respective scenes $\mathbf{x} \sim p_A(\mathbf{x} \mid \mathbf{v}, \mathbf{c})$ and $\tilde{\mathbf{x}} \sim p_A(\mathbf{x} \mid \tilde{\mathbf{v}}, \mathbf{c})$, for any homeomorphic, monotonic transformation $h_x \in \mathcal{H}_x$ such that $h_x(\mathbf{x}) = \tilde{\mathbf{x}}$, their exists another homeomorphic and monotonic transformation $h_v \in \mathcal{H}_v$ such that $\mathcal{H}_v \subseteq \mathcal{H}_x \subseteq \mathbb{R}^{\dim(\mathbf{x})}$ and $\mathbf{v} = h_v^{-1}\left(f_d^{-1}(h_x(\mathbf{x}); \mathbf{c})\right)$.

*Proof.* For a given view $\mathbf{v}$ and a mixing function $f_d$ that satisfy assumptions 2 and is piecewise affine, from theorem 2 we know the latent view representations are identifiable up to $\sim_s$ equivalence for a given aggregate content vector. Based on equation **??**, we know that $p(\mathbf{v})$ is expressed as GMM with a considered set of viewpoints, ideally learning each component for each viewpoint.

$$p(\mathbf{v}) = \sum_{v=1}^{|A|} \boldsymbol{\pi}_v \mathcal{N}(\mathbf{v}; \boldsymbol{\mu}_v, \boldsymbol{\sigma}_v)$$

Following similar arguments in Theorem 2 and Kivva et al. (2022), we can show that for a given content representation $\mathbf{c}$ the view distribution $p(\mathbf{v})$ is identifiable up to affine transformation. This means, for any two considered models $f_d, \tilde{f}_d$, such that $f_{d\sharp}(\mathcal{V}; \{\mathbf{c}\})$ and $\tilde{f}_{d\sharp}(\mathcal{V}; \{\mathbf{c}\})$ are equally distributed, then for any $\mathbf{x}^A \sim \mathcal{X}$ with the corresponding content representations given by $\mathbf{c}$ the views $\mathbf{v} = f_d^{-1}(\mathbf{x}^v; \mathbf{c})$, $\mathbf{v}' = \tilde{f}_d^{-1}(\mathbf{x}^v; \mathbf{c})$ are related in by an affine transformation $h(\mathbf{v}) = \mathbf{v}'$, results in:

$$\sum_{v=1}^{|A|} \boldsymbol{\pi}_v h_\sharp \left( \mathcal{N}(\mathbf{v}; \boldsymbol{\mu}_v, \boldsymbol{\sigma}_v^2) \right) \sim h_\sharp \left( \sum_{v=1}^{|A|} \boldsymbol{\pi}_v \mathcal{N}(\mathbf{v}; \boldsymbol{\mu}_v, \boldsymbol{\sigma}_v^2) \right), \tag{59}$$

Without loss of generality we can consider any function $f : \mathcal{C} \times \mathcal{V} \to \mathcal{X}$ is identifiable up to affine transformation, with this for given views $\mathbf{v}, \tilde{\mathbf{v}} \sim p(\mathbf{v})$ and for any object representations $\mathbf{c}$, the resulting scenes are sampled by distributions learned with mixing function $f$ is given by $\mathbf{x} \sim p_f(\mathbf{x} \mid \mathbf{c}, \mathbf{v}), \tilde{\mathbf{x}} \sim p_f(\mathbf{x} \mid \mathbf{c}, \tilde{\mathbf{v}})$. As previously established for some affine transformation $h$,

$$h(\mathbf{v}) = f^{-1}(\tilde{\mathbf{x}}; \mathbf{c}) \implies \mathbf{v} = h^{-1}\left( f^{-1}(\tilde{\mathbf{x}}; \mathbf{c}) \right) \tag{60}$$

Given $h_x(\mathbf{x}) = \tilde{\mathbf{x}}$, when combined with above equation we know $\mathbf{v} = h^{-1}\left( f^{-1}(\mathbf{x}; \mathbf{c}) \right), \tilde{\mathbf{v}} = h'^{-1}\left( f^{-1}(h_x(\mathbf{x}); \mathbf{c}) \right)$, for some invertible affine transformations $h$ and $h'$.

Given $h_x$ is homeomorphic and monotonic, and $f$ is piecewise linear, the inverse can be transferred resulting in $\tilde{\mathbf{v}} = h'^{-1}\left( \bar{h}_v(f^{-1}(\mathbf{x}; \mathbf{c})) \right)$, similarly we can also swap $h'^{-1}$ with $\bar{h}_v$, resulting in $\tilde{\mathbf{v}} = \bar{h}_v\left( h'^{-1}\left( f^{-1}(\mathbf{x}; \mathbf{c}) \right) \right)$. Additionally combining the results from theorem 2 and Kivva et al. (2022), we know that $h'^{-1} \circ h$ is an affine transformation $\bar{h}$. This results in:

$$\bar{h} = h'^{-1} \circ h \tag{61}$$

$$\implies \tilde{\mathbf{v}} = \left( \bar{h}_v \circ h \circ \bar{h} \right) \left( f^{-1}(\mathbf{x}; \mathbf{c}) \right) \tag{62}$$

$$\implies \tilde{\mathbf{v}} = h_v(\mathbf{v}) \tag{63}$$

Given affine transformation preserves monotonicity and homeomophism, the resulting transformation $h_v \in \mathcal{H}_v$ and $h_v \in \mathcal{H}_x$, concluding the proof.

$\square$

# F EXPERIMENTS

## F.1 SYNTHETIC DATASET RESULTS

Here, we illustrate visual results reflecting object binding in the case of view ambiguities. In figure 6 we demonstrate the results of MVPSA across 3 different views and compare them against PSA, and SA baselines. We additionally highlight some of the occluded regions which seem to better captured by our proposed model, which can be attributed to multi-view setting and the `sigmoid` mask. The spatial ambiguities in SA model misrepresents the blue dolphin in figure 6(a) as horse, which does not seem to be the case in the proposed model.

Additionally, we also illustrate the results from CLEVR-MV and GQN datasets in figures 7 and 8 respectively.

## F.2 MVMOVI RESULTS

Here, we discuss the results obtained from the proposed dataset. To reiterate, MVMOVI-C is a variant where fixed camera positions are maintained for all viewpoints across all scenes in the dataset. This setup helps assign a fixed type of viewpoint conditioning for all images captured from a particular camera.

The detection and binding quality of different models are illustrated in Table 2. From these results, we can clearly observe that while the model demonstrates similar binding capabilities, the identifiability of object representations is improved in our proposed model. This suggests that the use of fixed camera positions in MVMOVI-C enhances the consistency and quality of object representation learning, leading to better detection performance across different viewpoints.

Figure 9 showcases the object discovery capabilities of the MVPSA, PSA, and SA models, displayed from the top to the bottom row.

In the iteration of the MVMOVI-D dataset, we vary the camera position for each scene, making the dataset more dynamic and allowing for the potential violation of assumption 1 in certain cases.

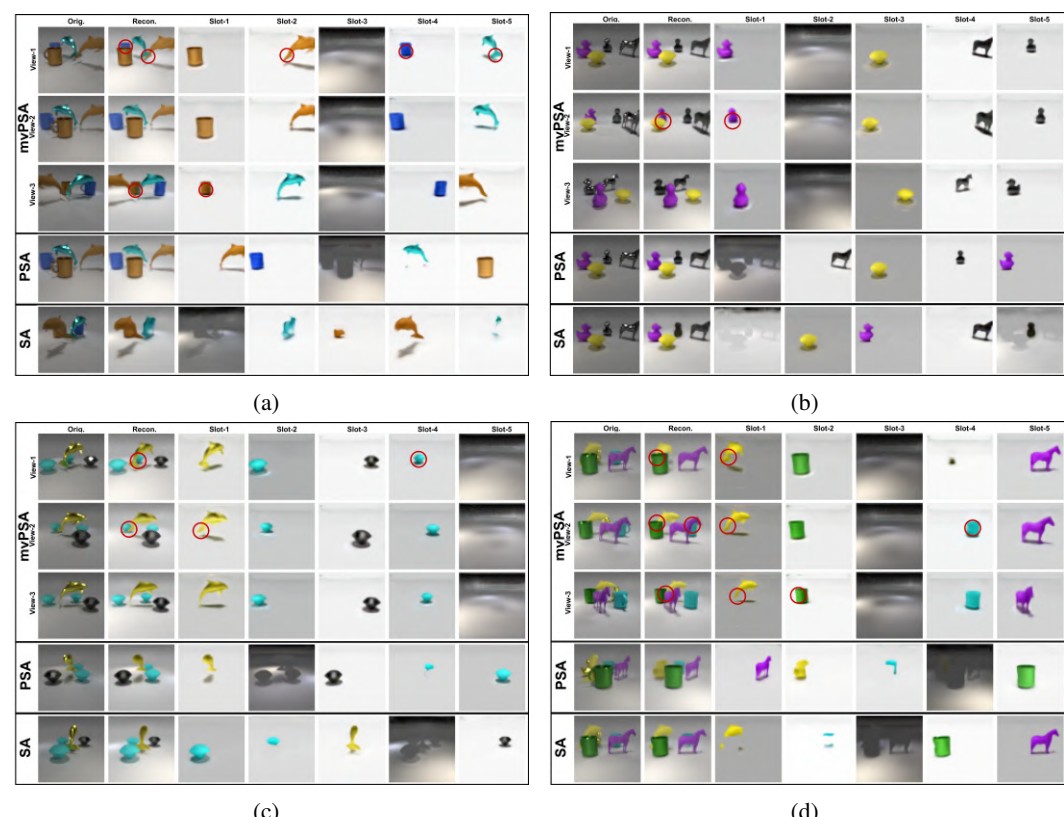

Figure 6: Visual illustrations of benchmark results on CLEVR-AUG dataset.

Table 3 presents the binding and identifiability results for both in-domain and out-of-domain data, following a similar analysis as in Table 2. We observe consistent trends and behaviors, suggesting that the impact of the assumption is minimal. A more detailed analysis of the assumption's effects will be left for future work.

Figure 10 similarly demonstrates the object discovery capabilities of the MVPSA, PSA, and SA models, arranged from top to bottom row.

Table 3: Identifiability and generalisability analysis on MV-MOVID dataset.

| METHOD | INDOMAIN ANALYSIS | | | | OUT OF DOMAIN | | | |
|---|---|---|---|---|---|---|---|---|
| | mBO ↑ | SMCC ↑ | INV-SMCC ↑ | MCC ↑ | mBO ↑ | SMCC ↑ | INV-SMCC ↑ | MCC ↑ |
| SA-MLP | 0.24 ± 0.031 | 0.44 ± 0.005 | - | 0.45 ± 0.007 | 0.24 ± 0.097 | 0.45 ± 0.008 | - | 0.49 ± 0.003 |
| PSA-MLP | 0.26 ± 0.022 | 0.44 ± 0.006 | - | 0.52 ± 0.017 | 0.25 ± 0.012 | 0.42 ± 0.006 | - | 0.50 ± 0.004 |
| MVPSA-MLP | 0.24 ± 0.099 | 0.48 ± 0.009 | 0.46 ± 0.054 | 0.57 ± 0.021 | 0.25 ± 0.011 | 0.48 ± 0.006 | 0.51 ± 0.021 | 0.55 ± 0.021 |
| SA-TRANSFORMER | 0.34 ± 0.017 | 0.40 ± 0.041 | - | 0.44 ± 0.005 | 0.34 ± 0.066 | 0.38 ± 0.031 | - | 0.44 ± 0.008 |
| PSA-TRANSFORMER | 0.37 ± 0.021 | 0.38 ± 0.007 | - | 0.46 ± 0.001 | 0.36 ± 0.024 | 0.36 ± 0.016 | - | 0.46 ± 0.007 |
| **MVPSA-TRANSFORMER** | 0.39 ± 0.016 | 0.46 ± 0.001 | 0.48 ± 0.001 | 0.54 ± 0.032 | 0.37 ± 0.051 | 0.46 ± 0.022 | 0.45 ± 0.010 | 0.54 ± 0.029 |

### F.3 OPTIMIZATION DETAILS

For training the MVPSA model on the large-scale MVMOVI datasets, we employ a gradual view addition approach. This is made possible by the model's inherent ability to handle an arbitrary number of viewpoints, as it is viewpoint-agnostic by design.

Initially, the model is trained using only single-view data, allowing it to focus on learning robust feature representations from a simpler setup. After 100,000 iterations, we progressively introduce

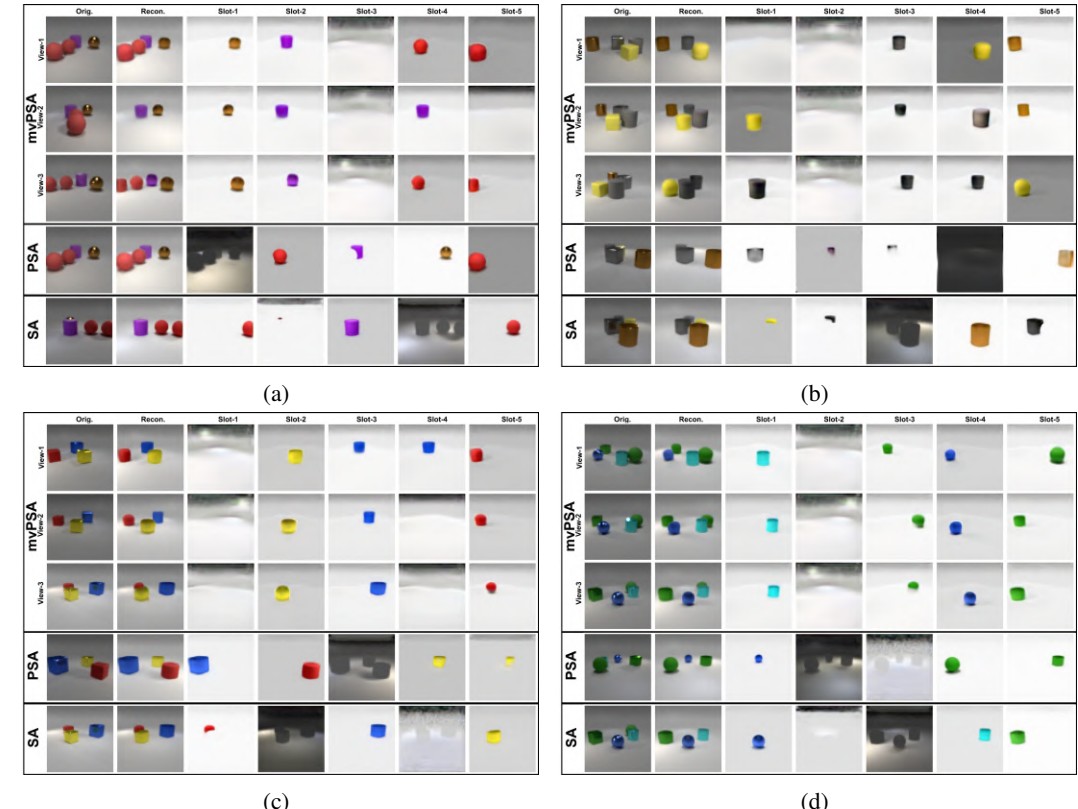

Figure 7: Visual illustrations of benchmark results on CLEVR-MV dataset.

additional viewpoints into the training pipeline. By doing so, the model incrementally learns to handle multi-view data without being overwhelmed by the complexity of multiple viewpoints from the start.

The primary motivation for this approach is to mitigate potential training uncertainties, particularly those caused by incorrect view matching in the aggregator module $g$. Gradually introducing views helps stabilize the training process, allowing the model to effectively bind and integrate information from different perspectives in later stages of training.

### F.4    HYPERPARAMETERS

In Table 4 we detail all the hyper-parameters used in our experiments. In the case of benchmark experiments, we use trainable CNN encoder as used in Locatello et al. (2020b); Kori et al. (2023), while in the case of proposed MVMOVI datasets we use DINO (Caron et al., 2021) encoder to extract image features and change our objective to reconstruct these features rather than the original image as proposed in Seitzer et al. (2022). For most of hyperparameters we use the values suggested by Locatello et al. (2020b); Seitzer et al. (2022), based on their ablation results.

### F.5    COMPUTATIONAL RESOURCES

We run all our experiments on a cluster with a Nvidia NVIDIA L40 48GB GPU cards. Our training usually takes between eight hours to a couple of days, depending on the model and the dataset. It is to be noted that speed might differ slightly with respect to the considered system and the background processes. All experimental scripts will be made available on GitHub at a later stage.

Table 4: Experimental details w.r.t datasets

| DATASETS(↓) | PARAMETERS | VALUES |
|---|---|---|
| CLEVR-MV, CLEVR-AUG | No. Layers | 4 |
| | No. Views | 10 |
| | No. Slots | 7 |
| | Training Epochs | 5000 |
| | Batch Size | 32 |
| | Optimizer | ADAM |
| | Learning Rate | 0.0002 |
| | Initial Slot $\boldsymbol{\mu}$ | $\mathcal{N}(0,1)$ |
| | Initial Slot $\boldsymbol{\sigma}$ | $\mathbb{I}$ |
| | Warmup Steps | 10000 |
| | Decoder | SPATIAL BROADCASTING-CNN |
| | $\mathbf{x}-$ likelihood | $\mathcal{N}(\boldsymbol{\mu}_x, \sigma_x^2 \mathbb{I})$ |
| GQN | No. Layers | 4 |
| | No. Views | 10 |
| | No. Slots | 4 |
| | Training Epochs | 5000 |
| | Batch Size | 64 |
| | Optimizer | ADAM |
| | Learning Rate | 0.0002 |
| | Initial Slot $\boldsymbol{\mu}$ | $\mathcal{N}(0,1)$ |
| | Initial Slot $\boldsymbol{\sigma}$ | $\mathbb{I}$ |
| | Warmup Steps | 10000 |
| | Decoder | SPATIAL BROADCASTING-CNN |
| | $\mathbf{x}-$ likelihood | $\mathcal{N}(\boldsymbol{\mu}_x, \sigma_x^2 \mathbb{I})$ |
| MVMOVI-C, MVMOVI-D | No. Layers | 4 |
| | No. Views | 5 |
| | No. Slots | 7 |
| | Training Epochs | 560 |
| | Batch Size | 64 |
| | Optimizer | ADAMW |
| | Learning Rate | 0.0002 |
| | Initial Slot $\boldsymbol{\mu}$ | $\mathcal{N}(0,1)$ |
| | Initial Slot $\boldsymbol{\sigma}$ | $\mathbb{I}$ |
| | Warmup Steps | 10000 |
| | Pretrained Encoder | DINO_VITB16 |
| | Decoder | MLP, TRANSFORMER |
| | $\mathbf{x}-$ likelihood | $\mathcal{N}(\boldsymbol{\mu}_x, \mathbb{I})$ |

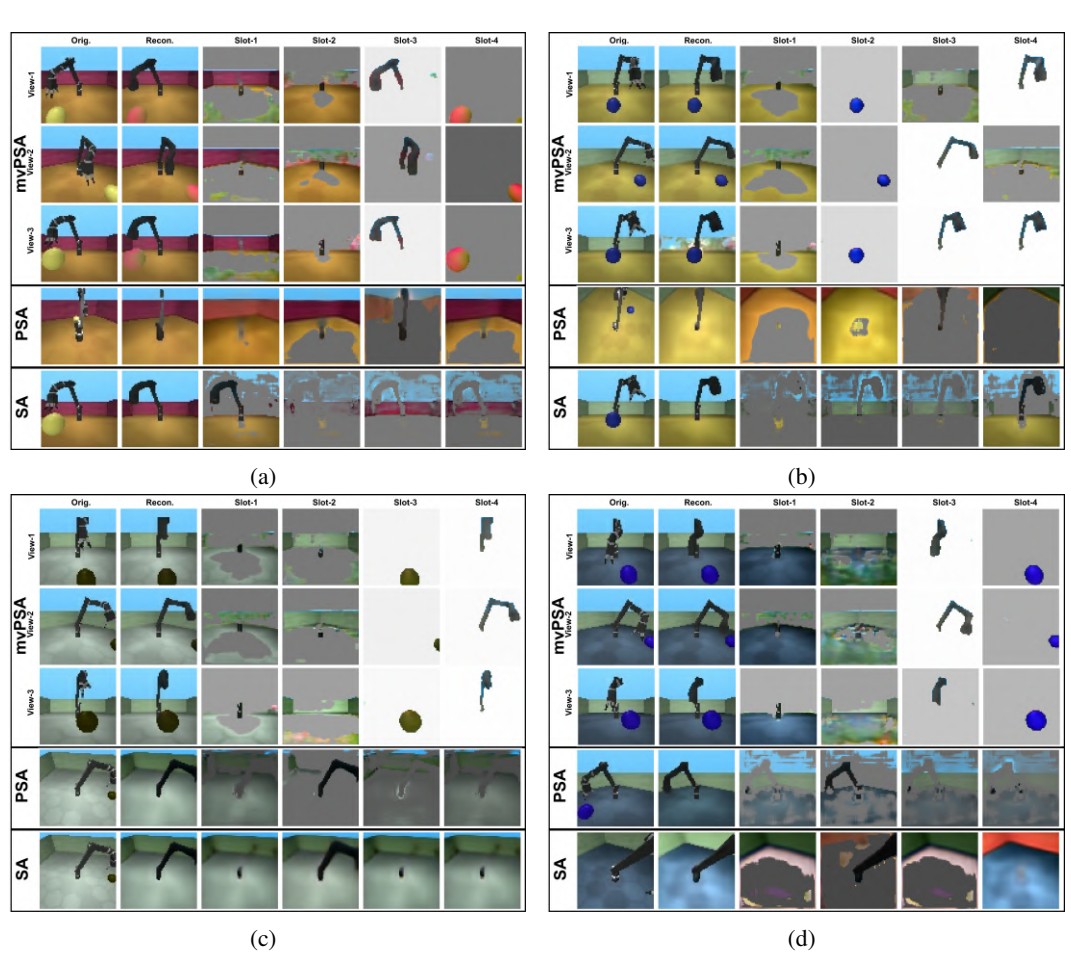

(a)

(b)

(c)

(d)

Figure 8: Visual illustrations of benchmark results on GQN dataset.

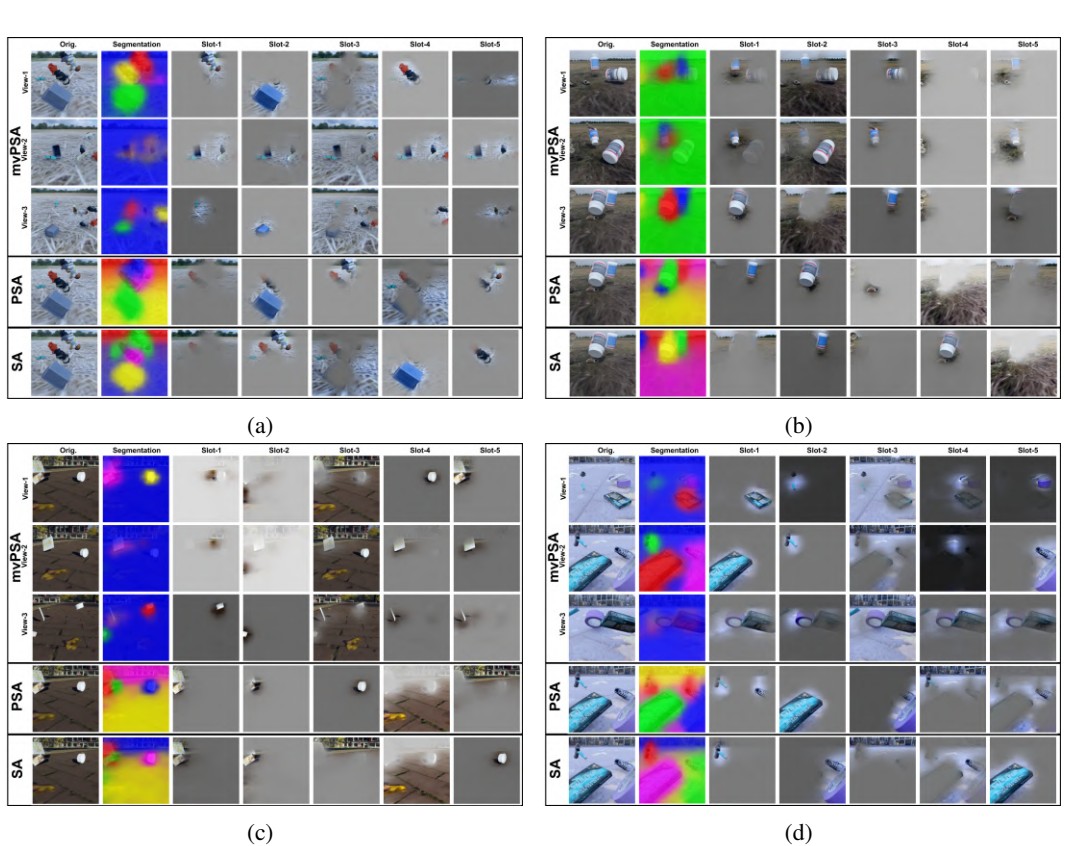

(a)

(b)

(c)

(d)

Figure 9: Visual illustrations of benchmark results on MVMOVI-C dataset.

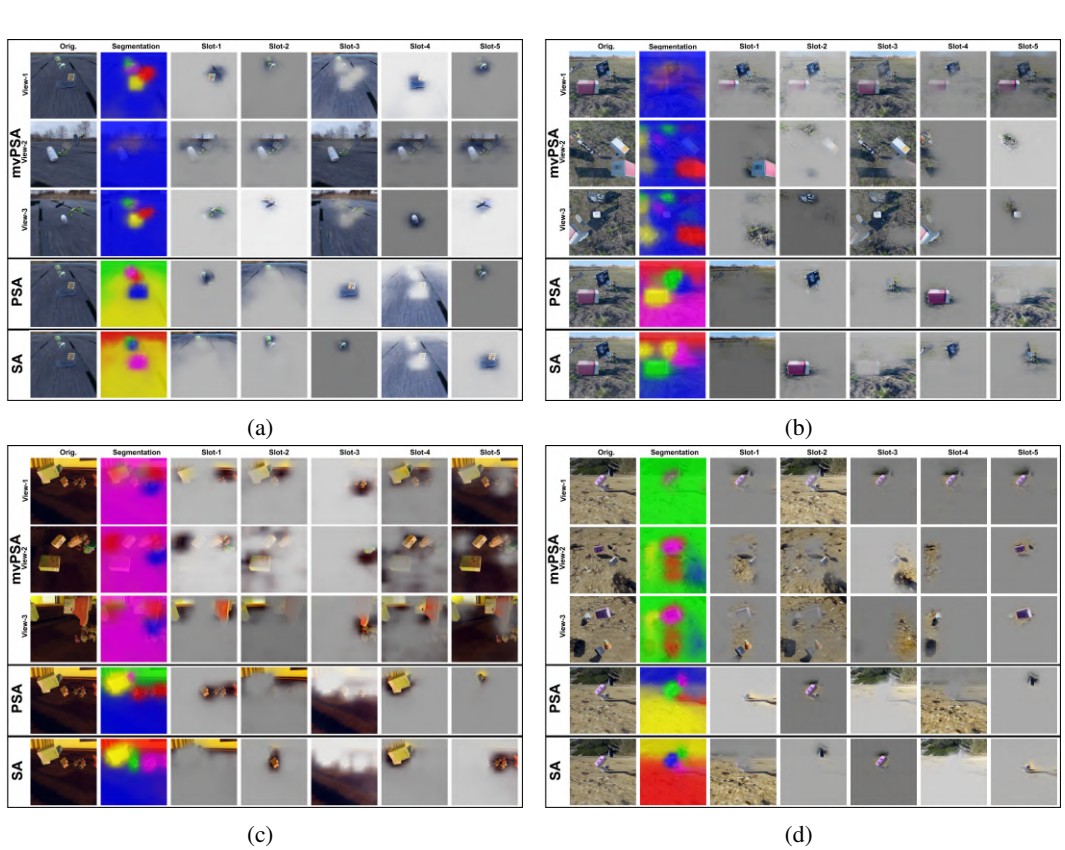

Figure 10: Visual illustrations of benchmark results on MVMOVI-D dataset.

