# OpenReview forum: "Multi-view Object-Centric Learning with Identifiable Representations"
_ICLR.cc/2025/Conference — Submitted to ICLR 2025_

### Official Review · Reviewer_jiM3 · 2024-10-23

**Soundness:** 2
**Presentation:** 3
**Contribution:** 2
**Rating:** 3
**Confidence:** 3

**Summary:**

The paper introduces a novel approach to learning object-centric representations that can handle challenges such as object occlusions. The proposed, theoretically guaranteed multi-view probabilistic method could jointly capture invariant content information and disentangled global viewpoint-level information. Experiments are conducted to justify the effectiveness of the proposed method.

**Strengths:**

- The paper addresses an important topic of multi-view object-centric representation learning.
- The theoretical justification of the proposed method is solid and easy to follow.

**Weaknesses:**

There are no concerns regarding the theoretical justification of this paper. However, due to the extremely poor empirical validation, the effectiveness of the proposed method cannot be justified. The authors only test their models on very simple scenes such as CLEVR or MOVi. In these scenarios, given multi-view images, it would be intuitive to leverage unposed 3D object-centric learning methods (e.g., [1]) or use few-view calibration methods like DUSt3R [2] to calibrate these scenes and then apply 3D object-centric learning methods (e.g., [3, 4, 5]) to extract per-object representation and obtain highly accurate object segmentation results. However, the authors did not provide a comparison between the proposed method and these previously established approaches, all of which fall under the category of direct inference. Furthermore, if per-scene optimization is permitted, one could even employ few-shot NeRF and Gaussian Splitting methods to first build a scene representation and then use segmentation models (e.g., [6]) to learn per-object representation and achieve scene segmentation.

Although the authors claim that the problem they tackle, namely multi-view object-centric learning, is a step forward, they overlook the internal difficulty of single-view object-centric learning, which is handling occlusion between objects. This paper sidesteps this challenge by utilizing multi-view information. While this is acceptable, as previous literature [1, 6] also considers multi-viewpoint setups, those works focus on real-world, in-the-wild scenes, unlike the simple synthetic scenes considered in this paper. Unsupervised object-centric representation learning on multi-view CLEVR or MOVi datasets is nearly a solved task [4, 5], so it is strongly recommended that the authors consider more complex datasets, such as Kitchen-Shiny [5], OCTScenes [8], or ScanNetv2 [9]. Thus, transitioning from single-view to multi-view settings while keeping the evaluation datasets the same might be considered a step backward.

[1] Sajjadi et al. Latent Neural Scene Representations from Unposed Imagery. In CVPR, 2023.

[2] Wang et al. DUSt3R: Geometric 3D Vision Made Easy. In CVPR, 2024.

[3] Yu et al. Unsupervised Discovery of Object Radiance Fields. In ICLR, 2023.

[4] Luo et al. Unsupervised Discovery of Object-Centric Neural Fields. arXiv, 2024.

[5] Liu et al. SlotLifter: Slot-guided Feature Lifting for Learning Object-centric Radiance Fields. In ECCV, 2024.

[6] Cen et al. Segment Any 3D Gaussians. arXiv, 2023.

[7] Sajjadi et al. Object Scene Representation Transformer. In NeurIPS, 2023.

[8] Huang et al. OCTScenes: A Versatile Real-World Dataset of Tabletop Scenes for Object-Centric Learning. arXiv, 2023.

[9] Dai et al. ScanNet: Richly-annotated 3D Reconstructions of Indoor Scenes. arXiv, 2017.

**Questions:**

See the weaknesses section. The reviewer believes the experiments of this paper cannot verify the effectiveness of the proposed method and strongly recommends the authors include experiments on more complicated, in-the-wild scenes, as mentioned above.

---

> ### Author Response · Authors · 2024-11-21
>
> We thank the reviewer for their feedback on our paper. We are pleased that the reviewers find our theoretical justification both robust and easy to follow.
>
> We also appreciate the reviewer for pointing us toward the related literature, which is orthogonal to the motivation in this work, however, we will incorporate discussions along this line of work. We would like to emphasise that the motivation and model used in our work are fundamentally different. Our focus is not on inverse rendering. Instead, the primary objective of this study is to establish the identifiability of object representations when an actual object is occluded. Specifically, we investigated the assumptions and scenarios under which such representations can be theoretically identified. These identifiability constraints are critical for real-world applications of OCL systems, particularly in handling more complex data and models.
>
> To validate our theoretical findings, we first conducted extensive experiments on controlled datasets, confirming our theoretical results empirically. Subsequently, we extended these demonstrations to unstructured image data. We stress that the use of synthetic datasets was essential for rigorously testing our identifiability hypotheses. To the best of our knowledge, prior to this work, there was no explanatory theory addressing the identifiability of multi-view object representations. While previous studies have demonstrated such results in single-view settings, they avoided spatial ambiguities, an aspect we specifically addressed in this work.

---

> ### Comment · Reviewer_jiM3 · 2024-11-21
> **Response to Author Rebuttal**
>
> Thank you to the authors for clarifying their contributions. I again acknowledge the theoretical contribution (in terms of the correctness of the proofs) of this paper. However, the task addressed by the paper—multi-view object-centric learning—is inherently a 3D problem. Thus, it could be approached from a 3D perspective and should be compared with existing 3D object-centric learning methods, also, on more complex datasets, as I have previously outlined.
>
> In my opinion, a theory-oriented paper should include sufficient experiments to validate its effectiveness. Currently, this paper demonstrates some theorems and applies the proposed algorithm to relatively simple CLEVR/MOVi datasets. This limited scope, in my view, does not sufficiently champion the paper’s acceptance.
>
> Therefore, I decide to maintain my score but reduce my confidence to 3. While I stand by my evaluation, I defer to the AC to determine the extent to which my recommendation should influence the final decision.

---

### Official Review · Reviewer_ARQH · 2024-11-02

**Soundness:** 2
**Presentation:** 2
**Contribution:** 2
**Rating:** 3
**Confidence:** 5

**Summary:**

This paper proposes a multi-view object-centric learning model (MVPSA) using identifiable representations, addressing challenges with occlusion and viewpoint ambiguities. MVPSA extends PSA into multi-view setting, aggregates content across views while separating viewpoint information. Theoretical guarantees for identifiability are provided, alongside extensive empirical results on synthetic datasets, demonstrating the model's ability to achieve identifiable object-centric representations without relying on additional viewpoint information.

**Strengths:**

1.	The paper introduces an object-centric learning method to address occlusion and spatial ambiguity in multi-view settings without relying on viewpoint conditioning, extending identifiability guarantees from single-view setups to multi-view settings.
2.	Both theoretical and empirical analyses are provided to support the model's identifiability claims. The extensive experiments conducted on multiple synthetic datasets, including the newly introduced MV-MOVI-C and MV-MOVI-D, validate the model's performance and robustness in various scenes.

**Weaknesses:**

1.	The approach is primarily based on the PSA framework, which lacks sufficient innovation, even though it extends the single-view method to a multi-view setting.
2.	The text in the figures is too small, making it difficult to read and comprehend the information presented.
3.	The related work section lacks recent methods for object-centric representation learning in multi-view settings, such as SIMONe[1] and OCLOC[2].


[1]	Kabra R, Zoran D, Erdogan G, et al. Simone: View-invariant, temporally-abstracted object representations via unsupervised video decomposition[J]. Advances in Neural Information Processing Systems, 2021, 34: 20146-20159.

[2]	Yuan J, Chen T, Shen Z, et al. Unsupervised Object-Centric Learning From Multiple Unspecified Viewpoints[J]. IEEE Transactions on Pattern Analysis and Machine Intelligence, 2024.

**Questions:**

1.	The proposed dataset has only 3,000 scenes, which raises questions about calling it "large-scale." Why it is considered large-scale compared to other datasets?
2.	I have tried extracting object representations from each viewpoint image using a single-view model, and I found that using Hungarian matching directly can lead to errors if an object is heavily occluded or completely hidden from one viewpoint. Does the model in this paper face this issue, and how might it be addressed?
3.	Judging from the visualization results in Appendix F, the segmentation and reconstruction effects of the proposed method are defective. Can the author analyze the reasons for the inferior results?
4.	Could the authors explain the purpose of the following proposed method in modeling viewpoint information separately?

---

> ### Author Response · Authors · 2024-11-21
>
> We thank the reviewer for their feedback on our paper, we are glad that the reviewers consider our theoretical and empirical results extensive.
>
> > **”The approach is primarily based on the PSA framework, which lacks sufficient innovation, even though it extends the single-view method to a multi-view setting.”**
> We respectfully disagree with this statement. While the proposed framework builds upon PSA, the theoretical findings, problem setup, and assumptions in this work differ significantly from PSA. The results concerning the identifiability of occluded objects and the identifiability of distributions across different subsets of view-point sufficient datasets are novel to this study.
> From a motivational standpoint, the issue of identifiability in the presence of occlusion is a well-recognized and important problem in OCL. To better reflect this, we will revise the introduction to clearly emphasize this aspect.
>
>
> > **”The related work section lacks recent methods for object-centric representation learning in multi-view settings, such as SIMONe[1] and OCLOC[2].”**
>
>
> Thanks for pointing them, we will include them in our related work, we are currently getting results on OCLOC for our considered datasets, we will be including them as an extra baseline model.
>
>
>
>
> > **”The proposed dataset has only 3,000 scenes, which raises questions about calling it "large-scale." Why it is considered large-scale compared to other datasets?”**
>
> The considered image has 3000 scenes, 5 views for each scene, and 24 images for each view making it 360,000 images in total. We consider this as large-scale because of the total number of involved images, diversity of objects and backgrounds considered and the image resolution. Considering all the baseline datasets, the proposed datasets are significantly large and diverse, adding more complexity and moving closer towards the real world setting.
>
> > **”I have tried extracting object representations from each viewpoint image using a single-view model, and I found that using Hungarian matching directly can lead to errors if an object is heavily occluded or completely hidden from one viewpoint. Does the model in this paper face this issue, and how might it be addressed?”**
>
> We address this issue with two steps:
>
> - Firstly, as detailed in Appendix F, we perform view warmup. Given the proposed model is invariant to the number of viewpoints, we first train the model with a single viewpoint data, for 10k iterations after that we introduce multiple viewpoint data. This allows us to learn a posterior pickup on the modes of the distribution which makes sure the Hungarian matching is performed on a meaningful representation preventing collapse while training.
>
> - Our aggregate content information is generated by considering convex combination of view-specific slot representation wrt mixing coefficient preventing these error cases to affect model training (please refer to intuition block at the beginning of page 5).

---

> > ### Author Response · Authors · 2024-11-21
> >
> > > **”Judging from the visualization results in Appendix F, the segmentation and reconstruction effects of the proposed method are defective. Can the author analyze the reasons for the inferior results?”**
> >
> > In this context, we would like to emphasize on two key points:
> >
> > - The primary objective of our work is to study the theoretical identifiability of object representations and the conditions under which this property holds, rather than to achieve state-of-the-art empirical results. Understanding the theoretical conditions for identifying representations of occluded objects is critical for real-world applications of OCL systems. These identifiability results are also foundational for scaling slot-based methods to high-dimensional images while maintaining correctness guarantees.
> >
> > - To verify our theoretical findings, we conducted comprehensive experiments on controlled datasets, demonstrating empirical validation for all theoretical results. We then extended these experiments to unstructured image data. Importantly, the synthetic datasets used were essential for rigorously testing our identifiability hypotheses. To our knowledge, prior to our work, there was no explanatory theory addressing the identifiability of multi-view object representations. While related results have been shown for single-view settings, such work has largely avoided addressing spatial ambiguities, a gap that our work explicitly addresses.
> >
> > Finally, regarding the segmentation column: this currently represents a continuous color map rather than a discretized segmentation map. To improve clarity, we will include additional columns showing the discretized segmentation maps in the results.
> >
> >
> > > **”Could the authors explain the purpose of the following proposed method in modeling viewpoint information separately?”**
> >
> > As detailed in L183: L187, we consider the viewpoint information is global, reflecting the property of the environment and not specific to each object. As described in the generative model in Figure 2(a) our data-generating process considers set of object-representations sampled from an aggregate GMM, while sample single view representation from view distribution composing them to create a desired scene.

---

> ### Comment · Reviewer_ARQH · 2024-11-24
> **Response to Author Rebuttal**
>
> Thank you to the authors for clarifying their contributions and innovations. I acknowledge the value of the theoretical findings presented in this work. However, I still believe that the approach to extending PSA to a multi-view setting lacks sufficient novelty, particularly in the method used to aggregate single-view results into multi-view results.
>
>
> A major weakness of the paper is the lack of comparisons and experiments involving recent related works, which makes it difficult to accurately evaluate the contributions and effectiveness of the proposed method.
> ﻿
>
> Furthermore, regarding the proposed datasets, I still do not find them sufficient to be classified as large-scale. Compared to the MOVi-C and MOVi-D datasets introduced in the original work, the new datasets are significantly limited in both the number of scenes and the number of objects per scene.

---

### Official Review · Reviewer_6KUk · 2024-11-02

**Soundness:** 2
**Presentation:** 2
**Contribution:** 2
**Rating:** 3
**Confidence:** 4

**Summary:**

This paper extends PSA to multi-view scenes and proposes a multi-view object-centric learning model (MVPSA) using identifiable representations. By aggregating all object representations in all multi-view scenes, identifiable representations across multi-view scenes are obtained. In addition, this paper proposes some theoretical proofs of recognizability. Experimental results on several synthetic datasets demonstrate that the model can achieve recognizable object-centric representations without relying on additional viewpoint information.

**Strengths:**

1.	This paper extends PSA from a single-view setting to a multi-view setting and provides certain theoretical guarantees.
2.	This paper verifies the performance and robustness of the proposed method in multi-view scenarios on multiple datasets.

**Weaknesses:**

1.	This paper seems to simply apply the PSA framework to multi-view scenarios through a Hungarian matching module and lacks innovation in both motivation and methodology.
2.	Some of the theoretical proofs and theorems are difficult to understand. It would be beneficial for the authors to provide more detailed explanations or examples to improve the readability of these proofs.
3.	Based on the visualization results presented in Appendix F, the segmentation and reconstruction performance of the proposed method appears to be less than expected, which raises concerns about the overall effectiveness and significance of the method.
4.	Some important works are missing for comparison (see Q5)

**Questions:**

1.	In Appendix C.2, it mentions that each scene has 5 viewpoints and 24 frames. What is the difference between viewpoints and frames?
2.	The formula in Line 213 of the main manuscript appears to have a typo. Are the brackets incomplete?
3.	The proposed method specifically models viewpoint information but does not seem to be used for object-centric representation learning. This makes one question the point of modeling perspective representation.
4.	How do the authors ensure that the Hungarian algorithm can correctly match all objects across viewpoints without falling into local optimality during training?
5.	Can the authors provide experimental results comparing the proposed method with other object-centric methods based on multi-view scenarios, such as SIMONe[1], OCLOC[2], etc.?

[1]	Kabra R, Zoran D, Erdogan G, et al. Simone: View-invariant, temporally-abstracted object representations via unsupervised video decomposition[J]. Advances in Neural Information Processing Systems, 2021, 34: 20146-20159.

[2]	Yuan J, Chen T, Shen Z, et al. Unsupervised Object-Centric Learning From Multiple Unspecified Viewpoints[J]. IEEE Transactions on Pattern Analysis and Machine Intelligence, 2024.

---

> ### Author Response · Authors · 2024-11-21
>
> We thank the reviewer for their feedback on our paper.
>
> > **”This paper seems to simply apply the PSA framework to multi-view scenarios through a Hungarian matching module and lacks innovation in both motivation and methodology.”**
>
> We respectfully disagree with this statement. While the proposed framework builds upon PSA, the theoretical findings, problem setup, and assumptions in this work differ significantly from PSA. The results concerning the identifiability of occluded objects and the identifiability of distributions across different subsets of view-point sufficient datasets are novel to this study.
> From a motivational standpoint, the issue of identifiability in the presence of occlusion is a well-recognized and important problem in OCL. To better reflect this, we will revise the introduction to clearly emphasize this aspect.
>
>
> > **”Some of the theoretical proofs and theorems are difficult to understand. It would be beneficial for the authors to provide more detailed explanations or examples to improve the readability of these proofs.”**
>
> We would like to highlight that we have provided intuitive explanations for each theorem, aiming to clarify the intent and implications of the results. If any part of the theorems or proofs remains unclear, we would greatly appreciate your feedback specifying which aspects require further clarification, and we will address them accordingly.
>
>
> > **”Based on the visualization results presented in Appendix F, the segmentation and reconstruction performance of the proposed method appears to be less than expected”**
>
> Could the reviewers kindly clarify what is meant by "expected" in this context? Is it in relation to the performance of the baseline models, their ground truth, or some other reference? We would also like to point out that Tables 2 and 3 tabulate the object discovery capabilities across all considered models for detailed comparison.
>
> Additionally, we wish to emphasize two key points:
>
> - The primary objective of our work is to study the theoretical identifiability of object representations and the conditions under which this property holds, rather than to achieve state-of-the-art empirical results. Understanding the theoretical conditions for identifying representations of occluded objects is critical for real-world applications of OCL systems. These identifiability results are also foundational for scaling slot-based methods to high-dimensional images while maintaining correctness guarantees.
>
> - To verify our theoretical findings, we conducted comprehensive experiments on controlled datasets, demonstrating empirical validation for all theoretical results. We then extended these experiments to unstructured image data. Importantly, the synthetic datasets used were essential for rigorously testing our identifiability hypotheses. To our knowledge, prior to our work, there was no explanatory theory addressing the identifiability of multi-view object representations. While related results have been shown for single-view settings, such work has largely avoided addressing spatial ambiguities, a gap that our work explicitly addresses.
>
> Finally, regarding the segmentation column: this currently represents a continuous color map rather than a discretized segmentation map. To improve clarity, we will include additional columns showing the discretized segmentation maps in the results.
>
>
>
> >  **”In Appendix C.2, it mentions that each scene has 5 viewpoints and 24 frames. What is the difference between viewpoints and frames?”**
>
> Apologies for the confusion, we will clarify this in the paper, the proposed dataset consists of 5 viewpoints and for each viewpoint, we collect 24 images reflecting dynamics of objects in the given environment (potentially creating a video snippet with 24 frames).
>
> > **”The formula in Line 213 of the main manuscript appears to have a typo. Are the brackets incomplete?”**
>
> Thanks for pointing it out, the inner closing bracket is not required. We will correct it.
> 	$\pi_k = \sum^{|A|}_v \tilde{\pi}^v_k / |A| $
>
> > **”The proposed method specifically models viewpoint information but does not seem to be used for object-centric representation learning. This makes one question the point of modeling perspective representation.”**
>
> As outlined in Lines 183–187, we treat the viewpoint information as global, representing a property of the environment rather than being specific to individual objects. As depicted in the generative model in Figure 2(a), our data-generating process involves sampling a set of object representations from an aggregate GMM. Subsequently, a single-view representation is sampled from the view distribution, and these are composed to create the desired scene.

---

> ### Author Response · Authors · 2024-11-21
>
> > **”How do the authors ensure that the Hungarian algorithm can correctly match all objects across viewpoints without falling into local optimality during training?”**
>
> As detailed in Appendix F, we perform view warmup. Given the proposed model is invariant to the number of viewpoints, we first train the model with a single viewpoint data, for 10k iterations after that we introduce multiple viewpoint data. This allows us to learn a posterior pickup on the modes of the distribution which makes sure the Hungarian matching is performed on a meaningful representation preventing collapse while training. We will detail this in the appendix even further.
>
> > **”Can the authors provide experimental results comparing the proposed method with other object-centric methods based on multi-view scenarios, such as SIMONe[1], OCLOC[2], etc.?”**
>
> Thanks for pointing to these interesting works, SIMONe corresponds to temporal settings and do not fall into the considered category of models as few of their implicit assumptions do not line up with the considered scenario. While, we are currently working on extracting results for OCLOC, which we will be including in our work.

---

> ### Comment · Reviewer_6KUk · 2024-11-26
>
> Thanks for addressing some of my concerns. However, I have not been entirely convinced with some of the authors' responses. For instance, the authors claim that by introducing the Hungarian algorithm to match representations of the same object across different viewpoints in images, the identifiability of occluded objects in the scene can be resolved. However, the representations of the same object with different levels of occlusion can be vastly different, potentially leading to inaccurate matches by the Hungarian algorithm. Furthermore, the submission lacks a detailed description of viewpoint representation learning, which makes me skeptical about the significance of the learned viewpoint representation. The authors are expected to provide visualization results of scene images generated through viewpoint interpolation and sampling to validate the effectiveness of the learned viewpoint representation. Lastly, in Figure 9 of Appendix F, the proposed method exhibits poor segmentation effects on multi-view scene images (such as only part of an object being segmented or multiple objects being segmented into a single slot), indicating that the proposed method performs inadequately in complex multi-view scenes and would be even more challenging to apply to real-world scenarios. Therefore, I tend to maintain my score.

---

### Official Review · Reviewer_M7A5 · 2024-11-03

**Soundness:** 3
**Presentation:** 3
**Contribution:** 3
**Rating:** 5
**Confidence:** 3

**Summary:**

This paper proposes a multi-view probabilistic approach that aggregates view-specific slots to capture invariant content information while learning disentangled global viewpoint features, advancing modular object-centric representations for human-like reasoning. The model addresses spatial ambiguities and provides theoretical guarantees for identifiable learning, outperforming prior single-view methods lacking theoretical support. The authors also present an identifiability analysis and extensive empirical validation, demonstrating strong performance on both benchmark datasets and newly designed large-scale multi-view datasets.

**Strengths:**

1. The paper provides identifiability guarantees specifically for multi-view scenarios, addressing a limitation in existing work that mainly focuses on single-view setups. This broadens the applicability of object-centric representations to more complex settings.
2. The proposed model is viewpoint-agnostic, requiring no additional view-conditioning information. This design improves the model's robustness and makes it adaptable across various viewing conditions without extra supervision.
3. The authors empirically validate the model’s scalability by testing it on large-scale datasets and using complex decoders, such as transformer-based decoders. This shows the model’s capacity to handle increasing data volume and decoder complexity effectively, which is crucial for practical applications.

**Weaknesses:**

1. The use of the Expectation-Maximization (EM) algorithm for fitting the GMM in Figure 2 is mentioned, but it’s unclear how this fits into the broader model architecture. Is this process repeated for each view, or is it a one-time initialization? Including a step-by-step outline or flowchart for the EM algorithm application might make this more transparent.
2. The description of viewpoint-specific slots and how they are represented as GMMs (Gaussian Mixture Models) in the multi-view context is technical but could benefit from more details. For instance, how are these slots initialized, and does the model assume independence between slots? Additionally, it would be helpful to discuss the computational implications of fitting GMMs for each view, especially in large-scale settings.
3. The explanation lacks details on the computational complexity of the aggregate GMM model, particularly as it scales with the number of views and components. It would be useful to discuss how computational tractability is ensured and any potential limitations in terms of scalability, given large datasets or high-dimensional view spaces.
4. Some expressions and details need further refinement to ensure clarity. For instance, in line 214, "...parameters described in 6" should refer to Equation 6 and should be written as "...parameters described in Equation 6."

**Questions:**

Please see weakness.

---

> ### Author Response · Authors · 2024-11-21
>
> We thank the reviewer for their effort and overall positive outlook on our paper. We are encouraged to read that our work  is perceived to broaden the object-centric learning to more complex real world and practical settings.
>
>
> > **"The use of the Expectation-Maximization (EM) algorithm for fitting the GMM in Figure 2 "**
>
> We thank the reviewer for pointing this out, we’ll include the textual description of how EM is applied. However we would like to point out that we have included an Algorithm in the appendix detailing all the steps involved in the proposed approach.
>
>
> > **”The description of viewpoint-specific slots and how they are represented as GMMs in the multi-view context is technical but could benefit from more details.”**
>
> Apologies for the confusion. We do include the initial slot distribution in the appendix, specifically in Table 4, and we will clarify this in the main text as well.
> Regarding computational complexity: mvPSA follows a complexity of $\mathcal{O}(VTNKD)$, which scales linearly with respect to the single-view PSA method. Here, $V$ represents the total number of views in the given batch, $T$ denotes the number of attention iterations, $N$ the number of input vectors, $K$ the number of slots, and $D$ the slot/input dimension. The additional complexity introduced by the aggregate object representation has a complexity of $\mathcal{O}(VK)$, which does not alter the dominant term. We will elaborate on this in the paper.
>
>
>
> > **”The explanation lacks details on the computational complexity of the aggregate GMM model, particularly as it scales with the number of views and components.”**
>
> It is important to note that we do not use the aggregate posterior during training; it is employed only for compositional generation. To generate the aggregate GMM, we select a subset of data points, extract their posterior representations, and merge them to create the parameters for the aggregate distribution. This process, however, can also be learned dynamically using approaches similar to VQ-VAEs [1] or GSD [2], where the parameters are estimated through mini-batch approximations, which we leave as a future work.
>
> > **”Some expressions and details need further refinement to ensure clarity. For instance, in line 214,”**
>
> Thanks for points this out, we will rephrase them for clarity.
>
> [1] Van Den Oord A, Vinyals O. Neural discrete representation learning. Advances in neural information processing systems. 2017;30.
>
> [2] Kori A, Locatello F, Ribeiro FD, Toni F, Glocker B. Grounded Object-Centric Learning. InThe Twelfth International Conference on Learning Representations 2023.

---

### Meta-Review · Area_Chair_VtLT · 2024-12-23

**Metareview:**

The paper provides theoretical and identifiability guarantees for multi-view object-centric learning. However, the reviewers have major concerns about the innovation of the paper, which was primarily based on the established PSA framework, the usage of the GMM model and its complexity, and the clarity of technical details. These concerns were not fully addressed after the discussion phase.

**Additional Comments On Reviewer Discussion:**

The reviewers partially agree with the authors' response but maintain their original scores despite acknowledging their efforts. The reviewers criticize its approach to multi-view object-centric learning for not adopting a 3D perspective and lacking novelty in aggregating single-view results. They also note insufficient experimental validation on complex datasets and a lack of detailed method comparisons.

---

### Decision · Program_Chairs · 2025-01-22

Reject